# The dorsomedial prefrontal cortex computes task-invariant relative subjective value for self and other

Matthew Piva[1]*, Kayla Velnoskey[2], Ruonan Jia[1], Amrita Nair[2], Ifat Levy[1,2,3,4], Steve WC Chang[1,2,4,5]

[1]Interdepartmental Neuroscience Program, Yale University, New Haven, United States; [2]Department of Psychology, Yale University, New Haven, United States; [3]Department of Comparative Medicine, Yale University School of Medicine, New Haven, United States; [4]Department of Neuroscience, Yale University School of Medicine, New Haven, United States; [5]Kavli Institute for Neuroscience, Yale University School of Medicine, New Haven, United States

**Abstract** Few studies have addressed the neural computations underlying decisions made for others despite the importance of this ubiquitous behavior. Using participant-specific behavioral modeling with univariate and multivariate fMRI approaches, we investigated the neural correlates of decision-making for self and other in two independent tasks, including intertemporal and risky choice. Modeling subjective valuation indicated that participants distinguished between themselves and others with dissimilar preferences. Activity in the dorsomedial prefrontal cortex (dmPFC) and ventromedial prefrontal cortex (vmPFC) was consistently modulated by relative subjective value. Multi-voxel pattern analysis indicated that activity in the dmPFC uniquely encoded relative subjective value and generalized across self and other and across both tasks. Furthermore, agent cross-decoding accuracy between self and other in the dmPFC was related to self-reported social attitudes. These findings indicate that the dmPFC emerges as a medial prefrontal node that utilizes a task-invariant mechanism for computing relative subjective value for self and other.
DOI: https://doi.org/10.7554/eLife.44939.001

*For correspondence:
matthew.piva@yale.edu

**Competing interests:** The authors declare that no competing interests exist.

## Introduction

Decision-making on behalf of other individuals is ubiquitous in daily life, from parents making decisions about the wellbeing of their children to financial advisors making economic decisions to maximize returns for their clients. A vital parameter in making decisions is the separation of the subjective value between the chosen and unchosen options, defined here as relative subjective value. When the values of two options are similarly preferred, relative subjective value is low, while relative subjective value is high when the values of two options are dissimilarly preferred. We encounter such considerations when we make decisions on behalf of others, much like we do when we make decisions for ourselves. It is therefore crucial to understand how such decisions are formulated in the brain when we make decisions for ourselves or for those around us.

Most previous studies have focused on subjective value in self-referenced decisions. These studies have identified neural correlates in the medial prefrontal cortex, including the ventromedial prefrontal cortex (vmPFC) (*Kable and Glimcher, 2007*; *Levy et al., 2010*; *McClure et al., 2004*) and dorsomedial prefrontal cortex (dmPFC), which is anatomically proximal or overlapping with multiple subregions including the dorsal anterior cingulate cortex (dACC) and pre-supplementary motor area (pre-SMA) (*Kolling et al., 2016*; *Kolling et al., 2012*). Other studies regarding self-referenced decision-making have focused instead on other decision-related processes, such as conflict, with most

indicating the importance of the dmPFC and dACC (*Botvinick et al., 2004*; *Botvinick et al., 2001*; *Braver et al., 2001*; *Lin et al., 2018*; *Venkatraman et al., 2009a*; *Venkatraman et al., 2009b*), and affect regulation, which is thought to be one of many roles ascribed to the vmPFC (*Delgado et al., 2016*). Together, these studies indicate clearly delineated functions of medial prefrontal subregions in value-based decision-making in self-referenced decisions.

On the other hand, existing studies of decision-making for others have reported divergent results. Early work using an intertemporal choice paradigm did not detect subjective value representation in the brain during decision-making for others (*Albrecht et al., 2011*). However, later work using a similar paradigm with behavioral modeling of subjective value determined a gradient from ventral to dorsal medial prefrontal cortex such that value for whoever a participant was making decisions at a given time was represented ventrally, while value for whoever a participant was not making decisions was represented dorsally (*Nicolle et al., 2012*). Still other studies using a prosocial learning framework have reported a categorically different ventral to dorsal gradient in the medial prefrontal cortex in which value for self is represented ventrally, while value for other is represented dorsally (*Sul et al., 2015*). This gradient was modulated by the social attitudes of participants, such that more prosocial individuals tended to display value representations for others more ventrally in the medial prefrontal cortex (*Sul et al., 2015*). Furthermore, a study implementing a computational model for describing altruistic choice based on a drift diffusion framework indicated that both self and other contributions to value calculation occur in the vmPFC (*Hutcherson et al., 2015*). Other studies using functional magnetic resonance imaging (fMRI) and electroencephalography (EEG) have also found evidence for shared neural signals of value across self and other (*Jenkins et al., 2008*; *Janowski et al., 2013*). Some studies have also used repetition suppression to show the impact of judging and learning about others' preferences on activity in the medial prefrontal cortex and ultimately on one's own preferences (*Garvert et al., 2015*; *Harris et al., 2018*). Finally, a recent study has suggested that the anterior dmPFC is involved in calculating normative value during social influence in decision-making (*Apps and Ramnani, 2017*). Together, the results of these studies seem to uniformly indicate the importance of various subregions of the medial prefrontal cortex in computing subjective value across both self and other. However, how individual subregions within the medial prefrontal cortex accomplish this calculation is still an area of active speculation, perhaps because most previous studies have relied on a single task and a single analytical method. To date, studies of value-based decision-making across self and other using multiple distinct experimental tasks and analytical techniques remain sparse.

To identify which brain areas amongst these medial prefrontal subregions compute relative subjective value from the perspective of self and other in a manner that is both invariant to different task demands as well as analytic approaches, we determined its neural correlates using fMRI in conjunction with both univariate and multivariate analytic techniques in a single unified study. To pinpoint task-invariant responses, we examined neural representations of relative subjective value for self and other under two independent sets of experiments, using either intertemporal or risky choice paradigms. Additionally, we varied the similarity of the other individual for whom the participant made choices. Finally, we examined whether the vmPFC and dmPFC, when analyzed using either univariate general linear model (GLM) or multivariate multi-voxel pattern analysis (MVPA) techniques, represent relative subjective value across self and other in both behavioral tasks regardless of the similarity of the other individual for whom participants made choices.

The 'common-currency' hypothesis (*Levy and Glimcher, 2012*; *Bartra et al., 2013*), a core theory in economics, could be useful in guiding expected results related to self and other representation of subjective value in our study. Although this theory has only recently been proposed in relation to decision-making studies exploring how people make decisions for other individuals (*Ruff and Fehr, 2014*; *Zaki et al., 2014*), explicit tests of whether the brain represents subjective value in an overlapping neural code for self and other and across different behavioral demands are currently absent from the literature. Whereas previous studies used only a single analytic technique and a single behavioral paradigm to explore social decision-making, we aimed to utilize a unique combination of analyses and experimental tasks to explicitly test for the first time whether value representation in any neural region is shared when people make decisions for themselves and others across varying behavioral contexts.

Our results indicated that relative subjective value signals were present in both the dmPFC and vmPFC across self and other as well as in both behavioral paradigms, such that dmPFC activity

negatively and vmPFC activity positively correlated with relative subjective value, respectively. Crucially, our results indicated that activity in the dmPFC, but not other medial prefrontal regions tested, was able to detectably encode relative subjective value, and this neural code retained information across self and other and displayed cross-task decodability. Together, these results present the first evidence for an overlapping neural code in the medial prefrontal cortex for self and other individuals that is remarkably consistent even across fundamentally different behavioral paradigms.

## Results

Two independent fMRI sessions were included, one in which participants performed an intertemporal choice task ($N = 20$) and one in which participants performed a risky choice task ($N = 21$). The intertemporal choice task (*Figure 1a*) involved participants making choices between lower amounts of money sooner or higher amounts of money later both for themselves ('Self' trials) and for another participant in the study ('Other' trials). Participants' baseline intertemporal choice preferences were assessed in a preliminary behavioral session outside the scanner (*Figure 1—figure supplement 1a, b*). The calculated discounting preferences were then used to pair high-discounting participants with low-discounting participants, and vice versa. Just prior to the fMRI session, participants learned about the person for whom they would be making choices in a separate learning task that involved guessing what the other person would choose before receiving feedback (*Figure 1—figure supplement 1c*). Minimum criterion to be included in the study was an accuracy of 80% (mean accuracy 89.3%, s.e.m. = 1.0%). For the intertemporal choice task, trial sets were designed individually for each participant via simulation to ensure that relative subjective value was uncorrelated between Self and Other trials. This was confirmed following completion of the study (for Self trials, mean correlation $r = 0.19$, s.e.m. = 0.10, $z = 1.57$, p=0.117, for Other trials, mean correlation $r = 0.15$, s.e.m. = 0.10, $z = 1.08$, p=0.279, Spearman correlation and Wilcoxon signed rank test). For the risky choice task (*Figure 1d*), participants made choices between lower amounts of money with higher probabilities or higher amounts of money with lower probabilities for themselves and for another participant in the study. However, while participants were told that they were making choices for another individual, the other participant in the risky choice task was fictional and based on the participants' own risk preferences as determined by a preliminary choice task completed online. Participants learned about the risk preferences of the fictional other participant in a learning task prior to scanning (*Figure 1—figure supplement 1d*), with criterion again set at 80% (mean accuracy 87.6%, s.e.m. = 1.2%).

### Behavioral modeling reveals distinct choice patterns for similar and dissimilar others

For the fMRI intertemporal choice sessions, behavior was modeled using hyperbolic decay functions with either one discounting parameter ($k$) across Self and Other trials or two discounting parameters with one for Self trials and one for Other trials. Low values for $k$ indicate that participants are willing to wait longer amounts of time, whereas high values for $k$ indicate that participants are less willing to wait to receive more money. Additionally, models were included that had either one noise parameter ($\beta$) for action selection across Self and Other trials or had two $\beta$ parameters, one for Self trials and one for Other trials. When participants were paired with another individual with very different discounting preferences, the hyperbolic model with two discounting parameters but one $\beta$ parameter fit best (*Figure 1b*; *Figure 1—source data 1*; mean BIC 66.8, s.e.m. = 3.4), with high discounters having higher discounting parameters for Self trials and low discounters having higher discounting parameters for Other trials (*Figure 1c*; high discounters $z = 2.80$, p=0.005, low discounters $z = 2.80$, p=0.005, Wilcoxon signed rank test).

Decision behavior in the risky choice task was modeled using prospect theory, in which risk attitudes are defined by $\alpha$. Low values for $\alpha$ indicate risk aversion, whereas high values for $\alpha$ indicate risk tolerance. We again tested separate models, with either a single value for $\alpha$ across Self and Other trials or two values for $\alpha$, one each for Self and Other trials. Variants of these models with either one or two $\beta$ action selection parameters were also included, for a total of four models. Partially diverging from the results of the intertemporal paradigm, the model with one value for $\alpha$ and one value for $\beta$ across self and other fit best (*Figure 1e*; *Figure 1—source data 1*; mean BIC 101.2, s.e.m. = 4.3) with no differences observed between $\alpha$ values for self and other in the model with two

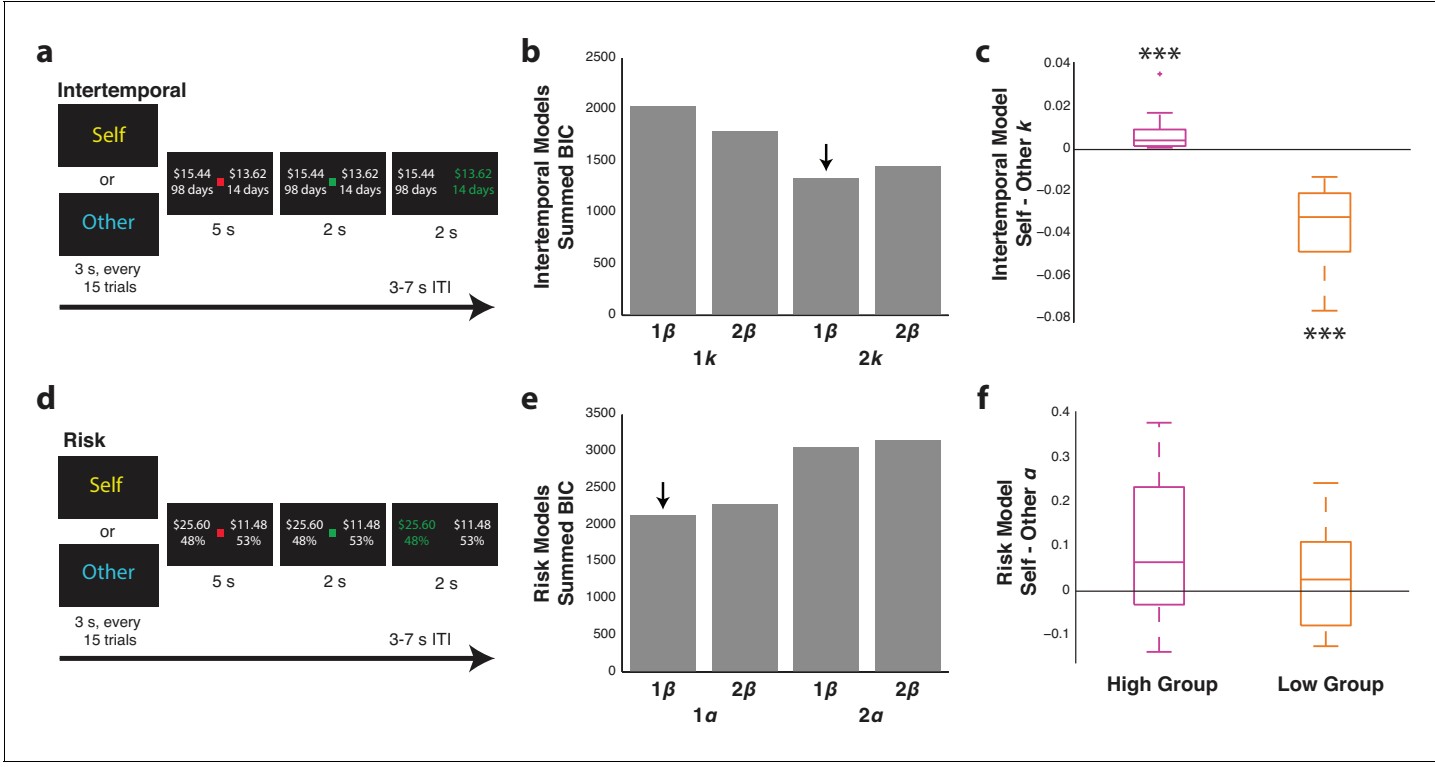

**Figure 1.** Participants differentiate between similar and dissimilar others in the intertemporal and risky choice tasks. (a,d) Intertemporal (top) and risky (bottom) choice task design. Every 15 trials, participants were presented with either 'Self' or 'Other' to indicate for whom they would be making choices in a given block. Each trial involved the presentation of two options, and participants were given the opportunity to indicate their choice either for themselves or for another participant in the study with a left or right button press. (b,e) Summed Bayesian information criterion (BIC) for the one and two discounting or risk parameter models for the intertemporal (top) and risky (bottom) choice tasks, respectively. For each of these models, variants with a common $\beta$ parameter or two $\beta$ parameters, one for Self and one for Other trials, were also included. Downward arrows indicate the best fitting model as determined via lowest BIC. (c,f) Self minus Other discounting parameters ($k$) or risk parameters ($\alpha$) for the intertemporal (top) and risky (bottom) choice two-parameter models, respectively, each with one $\beta$ parameter. For each analysis, participants were median-split into either high or low discounting or risk tolerance based on the value of their fitted $k$ or $\alpha$ parameters for Self trials. Data are plotted as box plots for each condition in which horizontal lines indicate median values, boxes indicate 25–75% interquartile range and whiskers indicate minimum and maximum values; data points outside 1.5x the interquartile range are shown separately as crosses. *** indicates p=0.005, Wilcoxon signed rank test. N = 20 participants for intertemporal choice with 10 high-discounting and 10 low-discounting individuals, N = 21 for risky choice with 11 high-risk tolerance and 10 low-risk tolerance individuals.

DOI: https://doi.org/10.7554/eLife.44939.002

The following source data and figure supplements are available for figure 1:

**Source data 1.** Bayes factor approximation for behavioral model fitting.

DOI: https://doi.org/10.7554/eLife.44939.006

**Figure supplement 1.** Behavioral results from the preliminary behavior-only intertemporal choice task as well as the intertemporal and risky choice learning tasks.

DOI: https://doi.org/10.7554/eLife.44939.003

**Figure supplement 2.** Choice behavior is more impacted by subjective value than individual trial variables.

DOI: https://doi.org/10.7554/eLife.44939.004

**Figure supplement 3.** Response times differ between high and low-relative subjective value trials but not for Self and Other trials for both behavioral paradigms.

DOI: https://doi.org/10.7554/eLife.44939.005

parameters for $\alpha$ and one $\beta$ parameter (*Figure 1f*; risk seeking $z$ = 1.48, p=0.139, risk averse $z$ = 1.00, p=0.328, Wilcoxon signed rank test). This indicated that participants viewed the other individual as someone with similar preferences. Together, these results indicated that participants considered similar or dissimilar preferences when making decisions for others. Furthermore, the fact that models with one $\beta$ parameter across Self and Other trials performed better than models with

two separate $\beta$ parameters for Self and Other trials in both tasks indicated that choice stochasticity did not substantially differ when participants made choices for self or other.

We next aimed to determine whether the subjective value of each option on a given trial calculated using the best fitting models for each paradigm accounted for choice behavior. To demonstrate that participants were taking subjective value into account when making choices, we calculated the proportion of trials in which participants chose the option with the higher subjective value, monetary value, or either lower delay for the intertemporal choice task or higher probability for the risky choice task. As anticipated, participants made choices according to subjective value significantly more than either monetary value or delay alone for both Self and Other trials in the intertemporal choice task (*Figure 1—figure supplement 2a*; compared to Self monetary, $z = 3.72$, $p<0.001$, Self delay, $z = 3.92$, $p<0.001$, Other monetary, $z = 3.85$, $p<0.001$, Other delay, $z = 4.01$, $p<0.001$, Wilcoxon signed rank test). Similarly, participants made choices according to subjective value significantly more than either monetary value or probability of reward for both Self and Other trials in the risky choice task (*Figure 1—figure supplement 2b*; compared to Self monetary, $z = 3.91$, $p<0.001$, Self probability, $z = 4.01$, $p<0.001$, Other monetary, $z = 4.01$, $p<0.001$, Other probability, $z = 4.01$, $p<0.001$, Wilcoxon signed rank test).

Finally, we examined response times in relation to the subjective value associated with each trial as well as whether participants were making decisions for themselves or another individual. For the subjective value analysis, we first calculated the relative subjective value of each trial by subtracting the value of the unchosen option from the value of the chosen option. We then median split trials into high and low relative subjective value. Response times were slower for trials with lower relative subjective value than trials with higher relative subjective value for both the intertemporal and risky choice paradigms (*Figure 1—figure supplement 3a*; intertemporal $z = 3.10$, $p=0.002$, risk $z = 2.38$, $p=0.017$, Wilcoxon signed rank test). However, no differences in response times were observed between Self trials and Other trials in either paradigm (*Figure 1—figure supplement 3b*; intertemporal $z = 1.01$, $p=0.313$, risk $z = 1.20$, $p=0.231$, Wilcoxon signed rank test). It is important to note that each trial included a mandatory waiting period of 5 s before participants were allowed to indicate their choice. While this may have diluted modulation of response times by task parameters, we still observed that response times were detectably modulated by subjective value.

## Decision-making for dissimilar but not similar others activates the 'social' brain

We first aimed to identify the neural correlates of decision-making from the perspective of self relative to other by performing a simple contrast of Other over Self trials, and vice versa. Notably, this analysis was agnostic to the subjective values assigned to the options of a given trial and was instead intended to merely identify whether our paradigm generally activated areas noted for importance in social cognition in previous studies. For the intertemporal choice paradigm in which participants were paired with others displaying dissimilar preferences, a contrast of other over self yielded significant activations in the ventromedial prefrontal cortex (vmPFC; peak Montreal Neurological Institute (MNI) coordinates [−8 42–20], $z_{peak} = 4.55$), the dorsomedial prefrontal cortex (dmPFC; coordinates [−4 52 36], $z_{peak} = 4.29$), and the temporal-parietal junction (TPJ; coordinates [−48–56 46], $z_{peak} = 4.42$; whole brain family-wise error (FWE) threshold at $p<0.05$, height threshold of $p<0.001$). All three of these clusters overlapped with a Neurosynth (*Yarkoni et al., 2011*) term search for 'social', indicating that these areas of the brain were associated with social cognition based on previous studies (*Figure 2*; *Figure 2—source data 1*). Interestingly, for the risky choice paradigm in which participants were paired with others displaying similar preferences, no clusters survived cluster correction at an FWE threshold of $p<0.05$ and a height threshold of $p<0.001$. No clusters were observed for the inverse contrast of Self over Other trials at this threshold for either paradigm. Decision-making for dissimilar others, compared to similar others, was thus largely responsible for driving activation of the 'social' brain in our dataset.

## Activity in the medial prefrontal cortex is differentiated by relative subjective value

In order to determine the effect of subjective value on prefrontal activity, we calculated relative subjective value, defined as the subjective value of the chosen option minus the subjective value of the

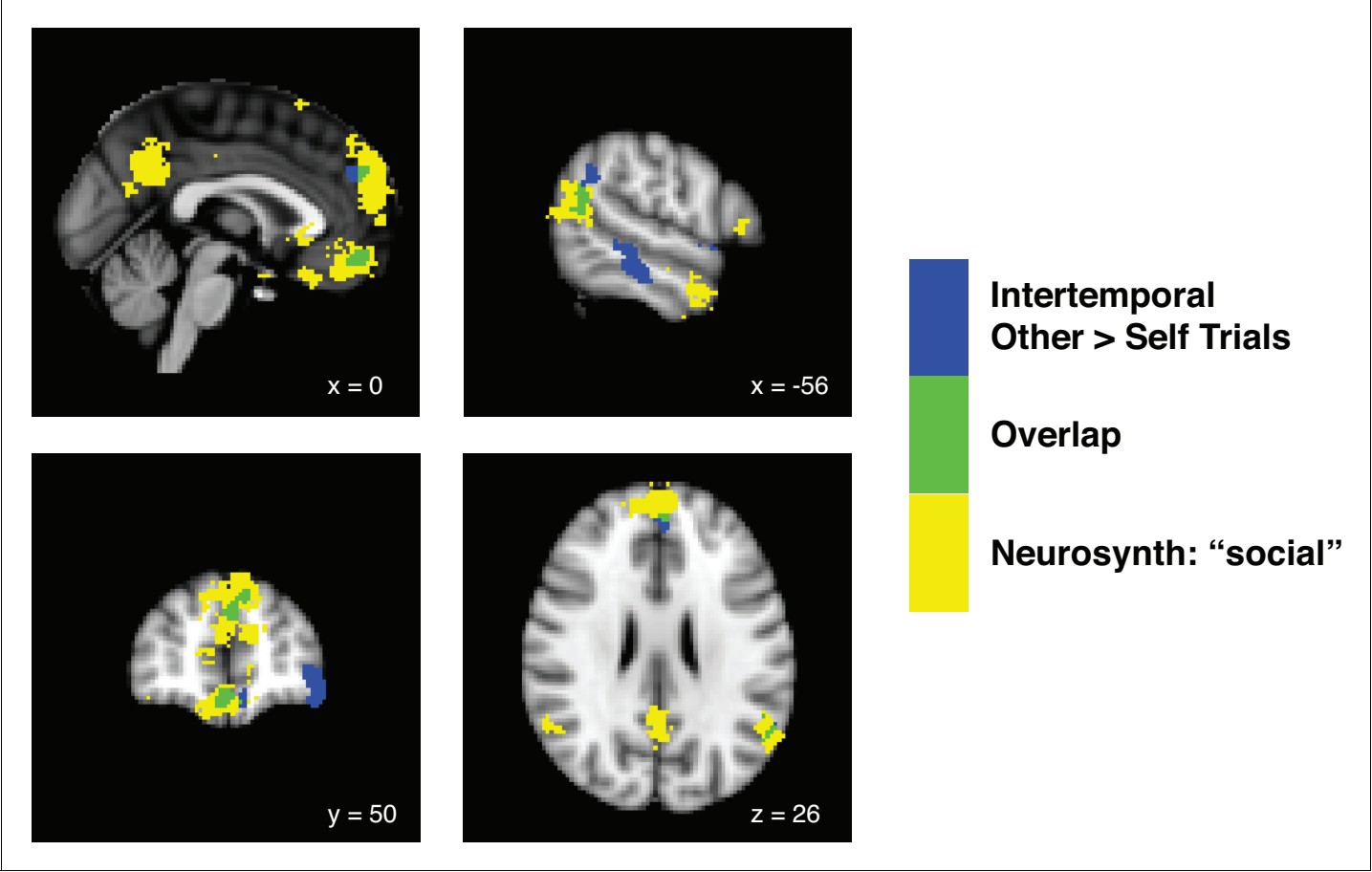

**Figure 2.** Decision-making for dissimilar others in the intertemporal choice paradigm drives activity in the 'social' brain. Statistical parametric maps showing the contrast of Other over Self trials during the intertemporal choice task (blue), a term search for 'social' in Neurosynth (yellow), and the overlap between the two analyses (green). We identified the neural correlates of decision-making from the perspective of other relative to self by performing a contrast of Other over Self trials, and vice versa. For the intertemporal choice paradigm in which participants were paired with others displaying dissimilar preferences, a contrast of other over self yielded significant activations in the ventromedial prefrontal cortex (vmPFC), the anterior portion of the dorsomedial prefrontal cortex (anterior dmPFC), and the temporal-parietal junction (TPJ) at a whole brain family-wise error (FWE) threshold of p<0.05, height threshold of p<0.001. All three of these clusters overlapped with a Neurosynth term search for 'social' (FDR-corrected to p<0.01), indicating that these clusters overlapped with brain areas associated with social cognition in previous studies. For the risky choice paradigm, in which participants were paired with others displaying similar preferences, no clusters survived even a lenient cluster correction at an FWE threshold of p<0.05 and a height threshold of p<0.01. No clusters were observed for the contrast of Self over Other trials at this threshold for either paradigm. N = 20 participants for intertemporal choice, N = 21 participants for risky choice.

DOI: https://doi.org/10.7554/eLife.44939.007

The following source data is available for figure 2:

**Source data 1.** Other over Self trials GLM contrast for the intertemporal choice paradigm and 'social' term search on Neurosynth.

DOI: https://doi.org/10.7554/eLife.44939.008

unchosen option. Using univariate GLM methods, we determined areas that corresponded to relative subjective value when pooling Self and Other trials together. For both the intertemporal and risky choice paradigms, activity in the vmPFC *positively* correlated with relative subjective value (intertemporal coordinates [0 46–4], $z_{peak}$ = 4.25; risk coordinates [0 42–8], $z_{peak}$ = 4.67), as did activity in the right temporal-parietal junction (TPJ; intertemporal coordinates [−58–40 24], $z_{peak}$ = 4.82; risk coordinates [−62–30 18], $z_{peak}$ = 5.09). Conversely, activity in the dmPFC *negatively* correlated with relative subjective value for both paradigms (intertemporal coordinates [8 22 48], $z_{peak}$ = 4.57; risk coordinates [0 14 52], $z_{peak}$ = 6.36; whole brain FWE threshold at p<0.05, height threshold of p<0.001). Relative subjective value clusters overlapped between paradigms in these three areas (*Figure 3a,b*; *Figure 3—source data 1*). These neural results held with the inclusion of response

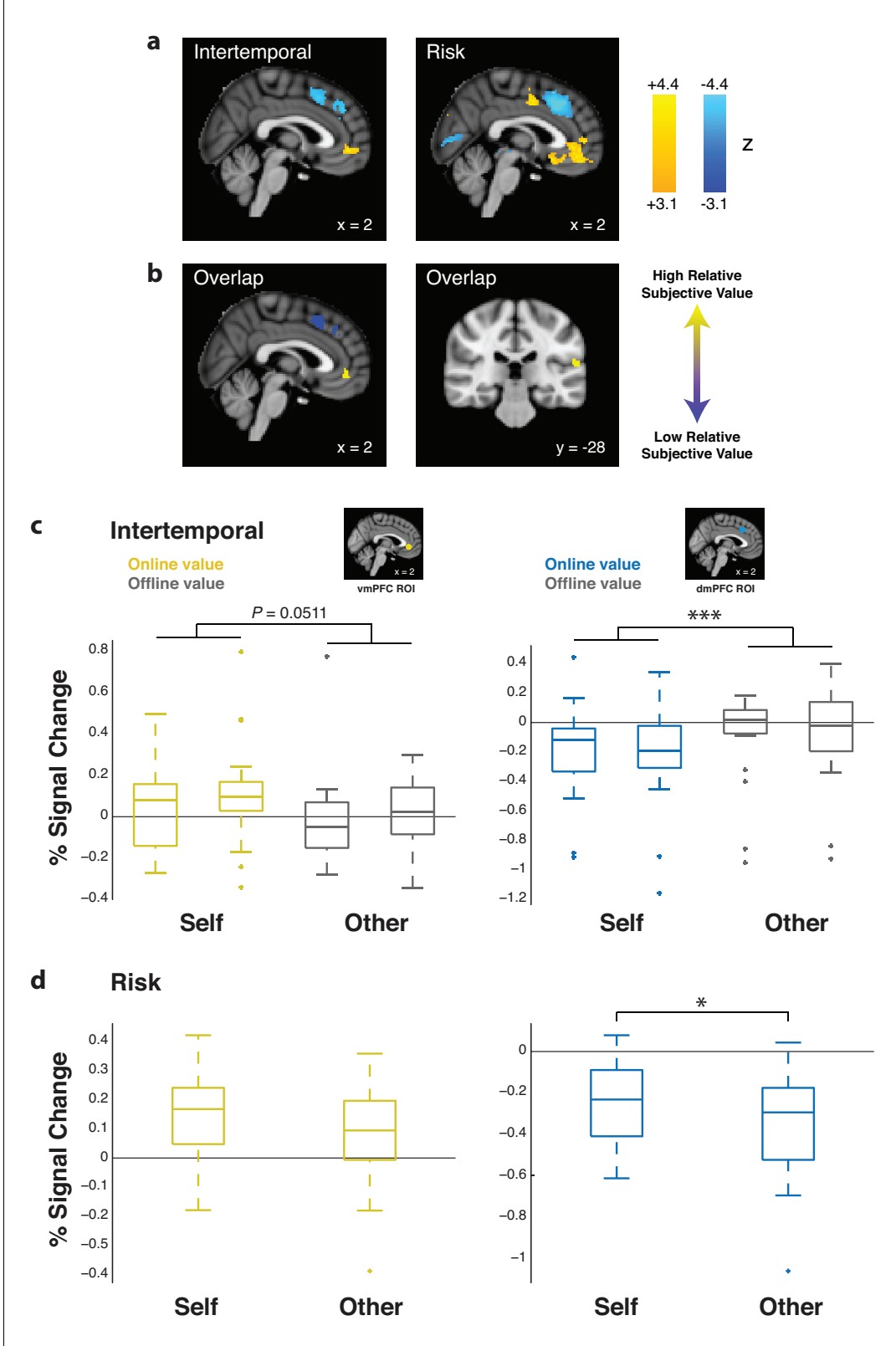

**Figure 3.** Ventromedial prefrontal cortex (vmPFC) and temporal-parietal junction (TPJ) activity is positively correlated with relative subjective value, while dorsomedial prefrontal cortex (dmPFC) activity is negatively correlated with relative subjective value. (a) Whole-brain statistical parametric map for the positive (yellow) and negative (blue) correlation with relative subjective value, thresholded at p<0.05 FWE-corrected, cluster-defining threshold p<0.001, for the intertemporal (left) and risky (right) choice paradigms (sagittal section, x = 2). (b) Simple overlap between the statistical maps for the *Figure 3 continued on next page*

*Figure 3 continued*

intertemporal and risky choice paradigms, including the positive (yellow) and negative (blue) correlation with relative subjective value (sagittal section, x = 2; coronal section, y = −28). (**c,d**) Extracted % signal changes for positive (left; yellow boxes; vmPFC ROI) and negative (right; blue boxes; dmPFC ROI) correlation with relative subjective value for the intertemporal (top) and risky (bottom) choice paradigms for Self and Other trials. Gray boxes indicate the corresponding % signal change for offline relative subjective value in the intertemporal choice task. Insets show 8 mm spherical ROIs for vmPFC (yellow) and dmPFC (blue) used to extract effect sizes. Data are plotted as box plots for each condition in which horizontal lines indicate median values, boxes indicate 25–75% interquartile range and whiskers indicate minimum and maximum values; data points outside 1.5x the interquartile range are shown separately as crosses. * indicates p<0.05, *** indicates p<0.005, all p>0.05 are explicitly stated. Significance was determined by comparison with either Wilcoxon signed rank or Friedman tests. N = 20 participants for intertemporal choice, N = 21 participants for risky choice.

DOI: https://doi.org/10.7554/eLife.44939.009

The following source data and figure supplements are available for figure 3:

**Source data 1.** Relative subjective value (RSV) GLM for Self and Other trials in the intertemporal and risky choice paradigms.

DOI: https://doi.org/10.7554/eLife.44939.013

**Source data 2.** GLM analysis of offline subjective value recapitulating the methods of *Nicolle et al. (2012)* in the intertemporal choice task.

DOI: https://doi.org/10.7554/eLife.44939.014

**Figure supplement 1.** Negative correlation with relative subjective value in Other over Self trials does not yield overlapping clusters between paradigms.

DOI: https://doi.org/10.7554/eLife.44939.010

**Figure supplement 2.** When recapitulating the methods of *Nicolle et al. (2012)*, evidence is found for representation of offline subjective value in the posterior dorsomedial prefrontal cortex (dmPFC) in the intertemporal choice task.

DOI: https://doi.org/10.7554/eLife.44939.011

**Figure supplement 3.** The absolute value of the chosen option is uncorrelated with activity in either the ventromedial prefrontal cortex (vmPFC) or dorsomedial prefrontal cortex (dmPFC).

DOI: https://doi.org/10.7554/eLife.44939.012

times as a nuisance regressor. Differences in the strength of relative subjective value representation between Self and Other trials were only observed when extracting effect sizes from the dmPFC in the risky choice task (*Figure 3c,d*; z = 2.24, p=0.025, Wilcoxon signed rank test). Additionally, contrasts of relative subjective value representation for Other over Self trials exhibited clusters that did not overlap across the intertemporal and risky choice paradigms (*Figure 3—figure supplement 1*; *Figure 3—source data 1*), indicating that differences in value representation for other versus self were not consistently observed across paradigms and could be due to specific task demands. In either case, these findings together suggest similar and overlapping univariate representation of relative subjective value for both Self and Other trials.

As a control, we extracted effect sizes from vmPFC (*Clithero and Rangel, 2014*) and dmPFC (*Venkatraman et al., 2009a*; *Venkatraman et al., 2009b*) ROIs for offline relative subjective value in the intertemporal choice paradigm. Offline relative subjective value was defined as the value associated with the agent for whom the participant was not making choices in a given trial. In other words, if one was making a choice for oneself, the discounting parameter associated with the other individual would be used to calculate relative subjective value, and vice versa. No offline relative subjective value clusters survived cluster correction at an FWE threshold of p<0.05 and a height threshold of p<0.001. Following extraction of effect sizes, no effects of offline value were found in the vmPFC or dmPFC for either Self or Other trials in the intertemporal choice paradigm (vmPFC Self trials z = 0.93, p=0.351, vmPFC Other trials z = 0.71, p=0.478, dmPFC Self trials z = 0.15, p=0.881, dmPFC Other trials z = 0.56, p=0.576, Wilcoxon signed rank test), and offline effects were found to be lower than online effects when directly compared (*Figure 3c*; vmPFC $\chi^2$ = 3.81, p=0.051, dmPFC $\chi^2$ = 13.49, p<0.001, Friedman test).

This control may at first seem to contradict an earlier report using a nearly identical experimental task that found that offline value was represented in the dmPFC (*Nicolle et al., 2012*). However, it is important to note that this previous study examined subjective value as a GLM contrast between neural representation of the subjective value of the chosen option over the neural representation of the subjective value of the unchosen option. In this study, we utilized only a single GLM regressor, defined as the difference between the chosen and the unchosen subjective value. When we recapitulated the analyses used in the previous report (*Nicolle et al., 2012*), neural representation of offline subjective value was indeed observed in the dmPFC when analyzing either Self and Other trials separately (Self trial coordinates [4 16 40], $z_{peak}$ = 5.47; Other trial coordinates [2 12 46], $z_{peak}$ = 6.08) or

together (*Figure 3—figure supplement 2*; *Figure 3—source data 2*; Self and Other trial coordinates [10 10 52], $z_{peak}$ = 5.84; whole brain FWE threshold at p<0.05, height threshold of p<0.001). Although this cluster was noted to be more posteriorly located than the cluster found in the original study (*Nicolle et al., 2012*), these results could be viewed as a replication of previous work.

Finally, we completed an ROI analysis for the vmPFC and dmPFC with the absolute subjective value of the chosen option. Notably, this analysis considered only the option chosen by participants on each trial without subtraction of the unchosen option. Neither ROI indicated any significant relationships (all p>0.05, Wilcoxon signed rank test), and all effects were observed to be significantly lower compared to relative subjective value effects when directly compared (*Figure 3—figure supplement 3*; intertemporal vmPFC $\chi^2$ = 8.01, p=0.005, intertemporal dmPFC $\chi^2$ = 12.52, p<0.001, risk vmPFC $\chi^2$ = 16.16, p<0.001, risk dmPFC $\chi^2$ = 40.10, p<0.001, Friedman test), indicating that medial prefrontal cortex activity in our paradigm was influenced by the relative comparison between the value of two options but not by the value of the chosen option itself.

## Activity in the dmPFC uniquely encodes relative subjective value for self and other

We utilized MVPA to determine whether vmPFC or dmPFC activity detectably encodes high versus low relative subjective value trials for self and other. Spherical ROIs (8 mm) for the vmPFC and dmPFC were based on previous studies (*Venkatraman et al., 2009a*; *Venkatraman et al., 2009b*; *Clithero and Rangel, 2014*), and one intermediate ROI (imPFC) was placed between the vmPFC and dmPFC in the medial prefrontal cortex (*Figure 4—source data 1*). Notably, classifiers trained using dmPFC activity were able to predict high versus low relative subjective value compared to an empirically derived null distribution, and this effect held over Self and Other trials and across paradigms (*Figure 4*; *Figure 4—source data 2*; all p<0.01 for dmPFC ROI for Self and Other trials from both paradigms, permutation test). The other ROIs tested failed to detectably encode relative subjective value in all cases (*Figure 4*; *Figure 4—source data 2*; all p>0.08, permutation test), with the exception of the imPFC in Self trials from the risky choice paradigm (*Figure 4b*; *Figure 4—source data 2*; p=0.031, permutation test). In the intertemporal paradigm, we were also able to test the classifier performance for the offline agent. Classifiers trained on activity in the dmPFC did not reach significance in decoding offline relative subjective value (Self trials p=0.107, Other trials p=0.057, permutation test), and there was a trend for higher accuracy for online relative to offline analyses when directly compared (*Figure 4—figure supplement 1*; $\chi^2$ = 2.65, p=0.104, Friedman test). While these results generally correspond to our univariate results, further work is needed to conclusively confirm the presence or absence of offline value signals in the dmPFC at the multivariate level. Together, these findings support the notion that the dmPFC is uniquely important in calculating relative subjective value across self and other compared to the other medial prefrontal structures examined.

We repeated these analyses for the vmPFC and dmPFC ROIs to determine whether activity in either area contained information related to the absolute subjective value of the chosen option in each trial. Corresponding to our univariate results, neither vmPFC nor dmPFC activity detectably encoded the value of the chosen option (all p>0.2 for both ROIs, permutation test), and classifiers trained using dmPFC activity were consistently able to decode relative subjective value more accurately when directly compared, with the exception of a strong trend for Other trials in the intertemporal choice task (*Figure 4—figure supplement 2*; Other trials intertemporal p=0.064, all other p<0.05, Wilcoxon signed rank test). Therefore, dmPFC activity contained information about the relative subjective valuation between options but not about the subjective value of the chosen option itself. To further confirm the ability of dmPFC activity to encode online relative subjective value, we used a dmPFC ROI derived from anatomical connectivity (*Neubert et al., 2015*) and observed that activity in the additional ROI detectably encoded relative subjective value (*Figure 4—figure supplement 3*; all p<0.005 for dmPFC ROI for Self and Other trials from both paradigms, permutation test), indicating that the effects we observed in the dmPFC were robust across different methods of ROI generation.

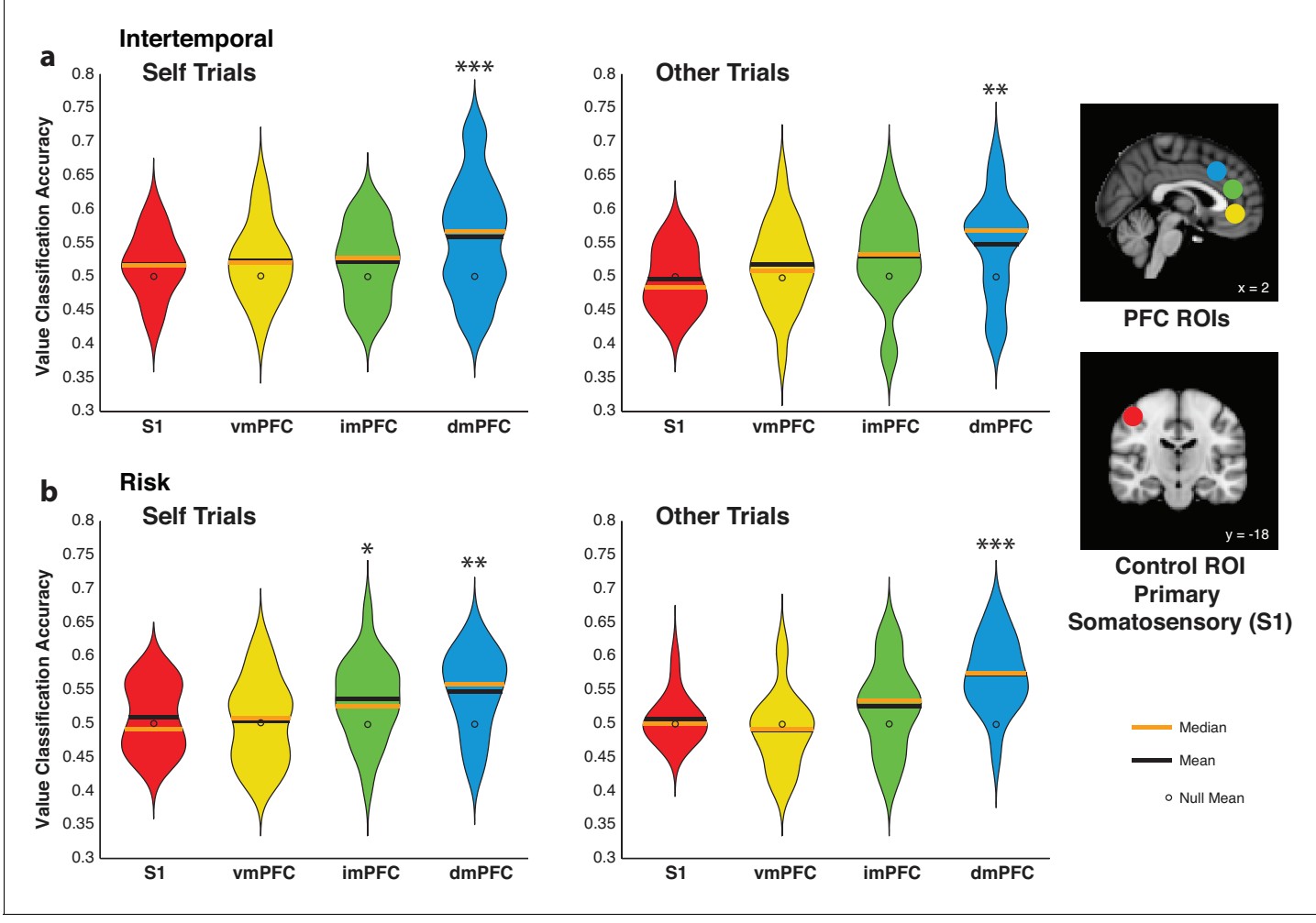

**Figure 4.** Activity in the dorsomedial prefrontal cortex (dmPFC) encodes relative subjective value for self and other. (a,b) Decoding accuracy of pattern classifiers trained on Self (left) and Other (right) trials in the intertemporal (top) and risky (bottom) choice paradigms for primary somatosensory cortex (S1), ventromedial prefrontal cortex (vmPFC), dmPFC, and an intermediate ROI (imPFC) placed between the vmPFC and dmPFC. Insets show anatomical location of 8 mm spherical ROIs. Plots indicate the full distribution of data using a kernel density estimation with a bandwidth of 0.025 applied to all plots. The orange horizontal line indicates the median of a distribution, the black horizontal line indicates the mean of a distribution, and the black dot indicates the mean of the corresponding empirically derived null distribution. * indicates p<0.05, ** indicates p<0.01, *** indicates p<0.005, permutation test. N = 20 participants for intertemporal choice, N = 21 participants for risky choice.

DOI: https://doi.org/10.7554/eLife.44939.015

The following source data and figure supplements are available for figure 4:

**Source data 1.** ROIs generated for multivariate analyses.

DOI: https://doi.org/10.7554/eLife.44939.019

**Source data 2.** MVPA significance for online relative subjective value analyses.

DOI: https://doi.org/10.7554/eLife.44939.020

**Figure supplement 1.** Dorsomedial prefrontal cortex (dmPFC) activity encodes online, but not offline, relative subjective value in the intertemporal choice task.

DOI: https://doi.org/10.7554/eLife.44939.016

**Figure supplement 2.** Dorsomedial prefrontal cortex (dmPFC) activity encodes relative subjective value, but not the absolute subjective value of the chosen option.

DOI: https://doi.org/10.7554/eLife.44939.017

**Figure supplement 3.** Activity in the anatomically defined dorsomedial prefrontal cortex (dmPFC) encodes relative subjective value.

DOI: https://doi.org/10.7554/eLife.44939.018

## The code for relative subjective value in the dmPFC generalizes over agents and contexts

In order to determine the similarity in the code for relative subjective value in the dmPFC for Self and Other trials, we trained pattern classifiers to decode relative subjective value from dmPFC data from Self trials and tested them on data from Other trials, and vice versa. The code for relative subjective value retained information over Self and Other trials in each paradigm, as cross-decoding accuracy was above the empirically derived null distribution for all comparisons (*Figure 5a,c*; *Figure 5—source data 1*; all p<0.05 for training on Self and testing on Other and vice versa for both paradigms, permutation test).

Moreover, we also tested whether the code for relative subjective value in the dmPFC could be generalized across the two behavioral paradigms. For this analysis, we examined a subset of participants (*N* = 10) who participated in both the intertemporal and risky choice fMRI sessions on separate days. We trained pattern classifiers to decode high versus low relative subjective value on either Self or Other trials from the intertemporal choice paradigm and then tested these classifiers on either the Self or Other trials from the risky choice paradigm, and vice versa. Indicating that the code for relative subjective value in the dmPFC was able to generalize across contexts, pattern classifiers trained on either Self or Other trials in the intertemporal paradigm were able to decode relative subjective value in either Self or Other trials in the risky paradigm (*Figure 5b*; *Figure 5—source data 1*; train Self intertemporal test Self risk p=0.055, all other p<0.05, permutation test). Correspondingly, pattern classifiers trained on either Self or Other trials in the risk paradigm were likewise able to decode relative subjective value in either Self or Other trials in the intertemporal paradigm (*Figure 5d*; *Figure 5—source data 1*; all p<0.05, permutation test). The ability of pattern classifiers to cross-decode relative subjective value similarly for Self and Other trials regardless of being trained on either Self or Other trials in the two distinct behavioral paradigms further underscores the retention of information in dmPFC activity across interpersonal reference frames and decision-making contexts.

## Encoding of relative subjective value in the dmPFC during Other trials reflects social attitudes

Our results so far demonstrate that the ability of classifiers trained using dmPFC activity to decode relative subjective value is consistent between Self and Other trials in both paradigms. We next determined whether the fidelity of dmPFC activity in encoding relative subjective value in Other compared to Self trials reflects task-independent self-reported social attitudes. Participants completed questionnaires relating to altruism, empathy, autism quotient, psychopathy, and social phobia. In order to collapse these metrics into a single behavioral score, we performed a principal component analysis and calculated the score of the first principal component for each participant. This component explained 44.8% of variance and loaded positively onto psychopathy, negatively onto altruism and empathy, and neutrally onto remaining measures (*Figure 6a*). These results indicated that this first principal component was primarily a measure associated with antisocial attitudes.

To determine the relationship between this measure and agent cross-decoding of relative subjective value across Self and Other trials, we averaged encoding of value when training on Self trials and testing on Other trials, and vice versa, in either the vmPFC or dmPFC. For participants who completed both the intertemporal and risky choice tasks, agent cross-decoding accuracy was averaged across both sessions for initial analyses. In hypothesizing the direction of such a correlation, there are two distinct possibilities. The first is that cross-decoding accuracy may negatively correlate with antisocial attitudes, indicating that high cross-decoding accuracy, and thus similar value coding across self and other, is associated with prosocial attitudes. Alternatively, cross-decoding accuracy may positively correlate with antisocial attitudes, indicating that low cross-decoding accuracy, and thus more differentiated value coding across self and other, is associated with prosocial attitudes. The latter possibility emphasizes that an ability to differentiate value coding across self and other is critical for expressing preference toward another individual's benefit, akin to the importance of mentalizing about others' preferences.

There was no correlation observed between agent cross-decoding accuracy in the vmPFC and our antisocial personality measure (*Figure 6b*; *r* = −0.08, p=0.895, Spearman correlation). By contrast, there was a positive correlation observed in the dmPFC (*Figure 6c*; *r* = 0.40, p=0.025,

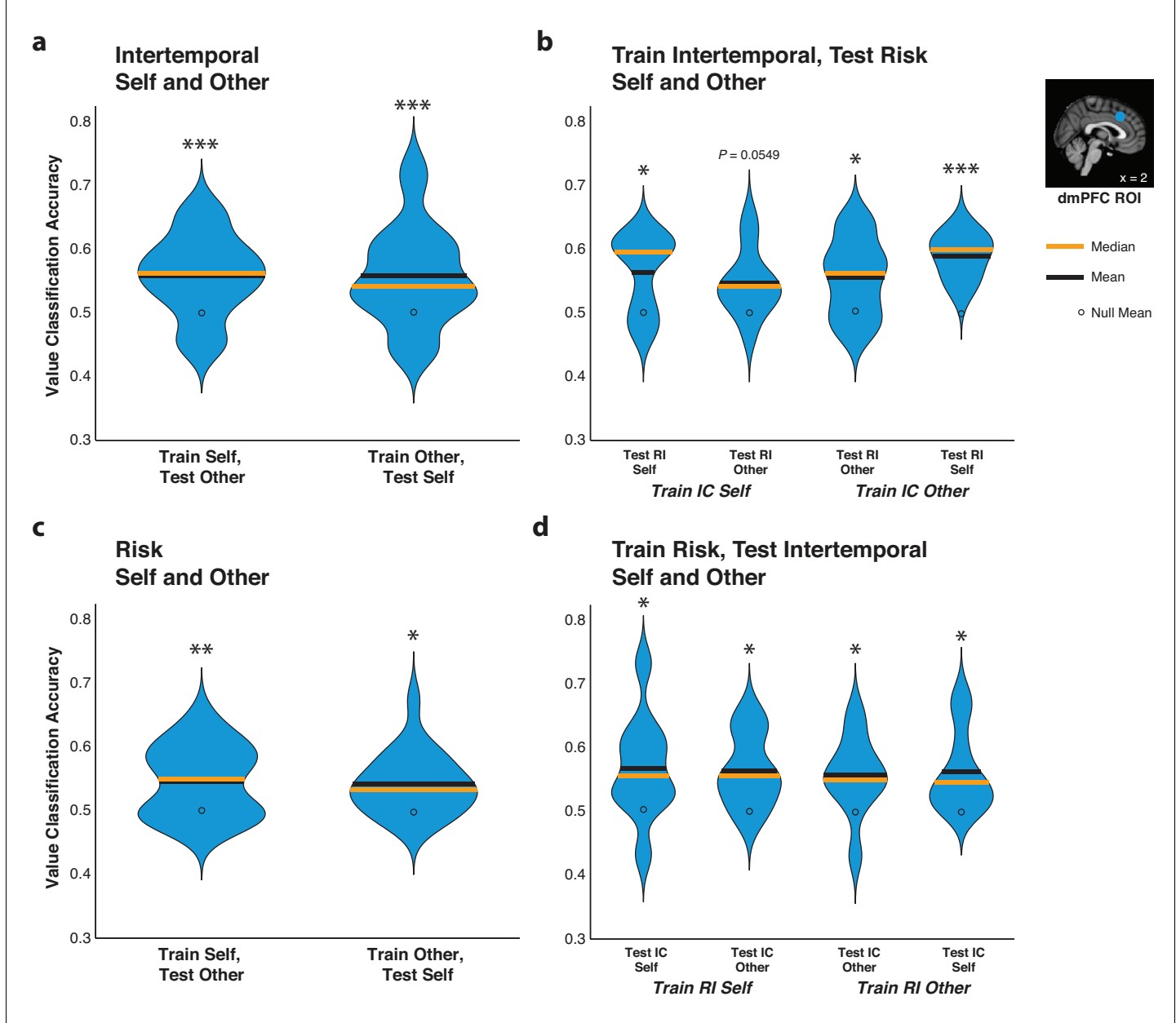

**Figure 5.** The code for relative subjective value in the dorsomedial prefrontal cortex (dmPFC) retains information over agents and contexts. (a,c) Agency generalizability. Decoding accuracy of pattern classifiers trained on dmPFC data from Self trials and tested on dmPFC data from Self or Other trials as well as pattern classifiers trained on dmPFC data from Other trials and tested on dmPFC data from Self or Other trials for both the intertemporal (top) and risky (bottom) choice paradigms. (b,d) Paradigm generalizability. Decoding accuracy of pattern classifiers trained on dmPFC data from Self or Other trials in the intertemporal choice (IC) task and tested on dmPFC data from Self or Other trials in the risky choice (RI) task (top) or tested on data from the RI task and tested on data from the IC task (bottom). Plots indicate the full distribution of data using a kernel density estimation with a bandwidth of 0.025 applied to all plots. The orange horizontal line indicates the median of a distribution, the black horizontal line indicates the mean of a distribution, and the black dot indicates the mean of the corresponding empirically derived null distribution. Inset shows anatomical location of the 8 mm spherical dmPFC ROI. * indicates p<0.05, ** indicates p<0.01, *** indicates p<0.005, all p>0.05 are explicitly stated, permutation test. $N = 20$ participants for intertemporal choice, $N = 21$ participants for risky choice, $N = 10$ for (c) and (d) in which data were trained on intertemporal and tested on risk, and vice versa.

DOI: https://doi.org/10.7554/eLife.44939.021

The following source data is available for figure 5:

**Source data 1.** MVPA significance for online relative subjective value analyses across agents and tasks.

DOI: https://doi.org/10.7554/eLife.44939.022

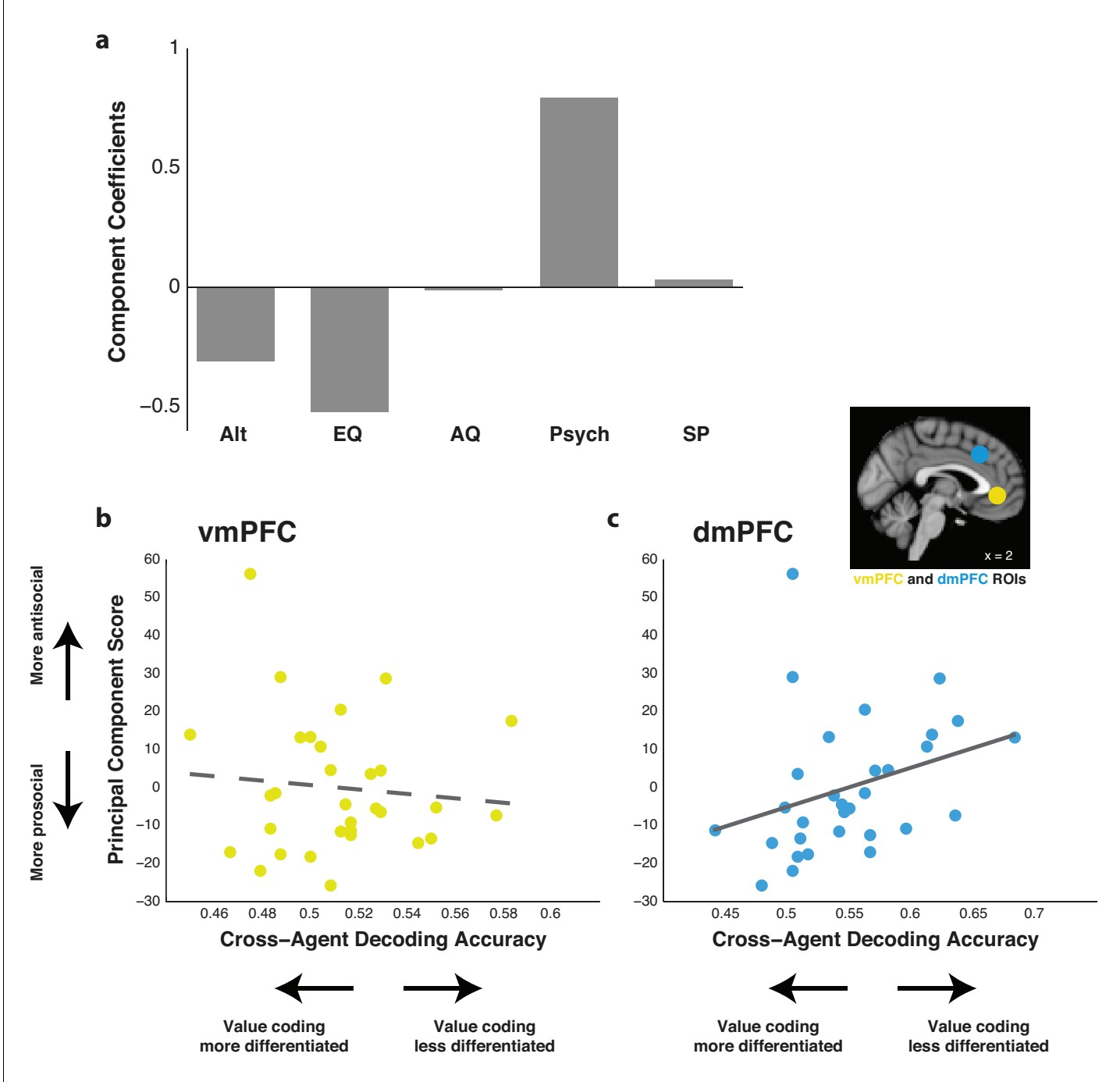

**Figure 6.** Agent cross-decoding of relative subjective value classifiers trained on dorsomedial prefrontal cortex (dmPFC) activity correlates with social attitudes. (a) Component coefficients of the first principal component score across self-reported measures for altruism (Alt), empathy quotient (EQ), autism quotient (AQ), psychopathy (Psych), and social phobia (SP). (b,c) Correlation between the first principal component score and the agent cross-decoding accuracy of relative subjective value of classifiers trained on data from a given ROI. For participants who completed both intertemporal and risky choice sessions, cross-decoding accuracy was averaged. Higher principal component scores indicate higher antisocial attitudes. Higher cross-decoding accuracy indicates that participants' value codes for Self and Other trials were less differentiated. The ventromedial prefrontal cortex (vmPFC; b) did not yield a significant correlation, noted by the dotted trend line. However, the dmPFC (c) yielded a significant correlation, noted by the solid line ($r = 0.403$, $p=0.025$, Spearman's correlation with significance verified via permutation test). Inset shows the anatomical location of the 8 mm spherical ROIs for the vmPFC (yellow) and dmPFC (blue). $N = 31$ participants for all correlations.

DOI: https://doi.org/10.7554/eLife.44939.023

The following figure supplements are available for figure 6:

*Figure 6 continued on next page*

*Figure 6 continued*

**Figure supplement 1.** Social attitudes correlate with cross-decoding accuracy for both training on Self trials and testing on Other trials, and vice versa, in the dorsomedial prefrontal cortex (dmPFC).
DOI: https://doi.org/10.7554/eLife.44939.024
**Figure supplement 2.** Agent cross-decoding of relative subjective value classifiers trained on dorsomedial prefrontal cortex (dmPFC) activity correlates with social attitudes separately for both the intertemporal and risky choice tasks.
DOI: https://doi.org/10.7554/eLife.44939.025

Spearman correlation; p<0.05 confirmed via permutation test), indicating that participants with greater agent cross-decoding in the dmPFC had higher measures of antisocial attitudes. To confirm this relationship, we examined the correlation between social attitude scores and the individual accuracy when training on Self trials and testing on Other trials, and vice versa, in the vmPFC and dmPFC. Again, no correlation was observed in the vmPFC for either analysis (*Figure 6—figure supplement 1a*; train Self test Other $r = 0.03$, p=0.880, train Other test Self $r = -0.08$, p=0.653, Spearman correlation). Examination of the dmPFC yielded either a significant or strong trend toward a positive correlation (*Figure 6—figure supplement 1b*; train Self test Other $r = 0.453$, p=0.011, train Other test Self $r = 0.341$, p=0.061, Spearman correlation), confirming the cross-decoding relationship observed in the dmPFC but not in the vmPFC. We further performed this analysis separately on data from the intertemporal and risky choice tasks. As in other analyses, no correlation was observed in the vmPFC for either paradigm (*Figure 6—figure supplement 2a*; intertemporal $r = -0.147$, p=0.537, risk $r = -0.040$, p=0.862, Spearman correlation). However, even with diminished statistical power from reduced sample size, we still observed a trend toward a positive correlation in both tasks in the dmPFC (*Figure 6—figure supplement 2b*; intertemporal $r = 0.415$, p=0.070, risk $r = 0.423$, p=0.056, Spearman correlation). These results indicate that the observed relationship was not driven by the similarity or dissimilarity of the other participant or by specific task demands. As mentioned above, since agent cross-decoding is primarily a measure of how similar value coding is for Self and Other trials in the dmPFC, these results indicate that participants who report more antisocial attitudes tend to have value coding that is less differentiated when making decisions for themselves as opposed to other individuals in our experimental paradigms.

## Discussion

While making decisions for others is vital in daily life, previous studies examining the neural correlates of subjective value during decision-making for others have yielded mixed and sometimes conflicting results. These divergent findings could be the result of utilizing tasks with different associated cognitive demands and behavioral models. To address this issue, we utilized two paradigms, intertemporal and risky choice, as well as two different variations of other, those that had similar versus dissimilar preferences to each participant. When using relative subjective value, defined as the subjective value of the chosen minus unchosen option, as a behavioral regressor, both univariate and multivariate analytic techniques yielded remarkably consistent results across decision-making for self and other as well as across intertemporal and risky choice paradigms. In all univariate cases, the vmPFC positively correlated with relative subjective value, while the dmPFC negatively correlated with relative subjective value. When taking a multivariate approach, the dmPFC, but not other medial prefrontal areas, contained detectable information concerning the relative subjective value of each trial. These results were consistent across self and other, and the code for relative subjective value retained information across self and other as well as across behavioral paradigms. Finally, a relationship between self-reported social attitudes and agent cross-decoding accuracy was observed. These results are particularly novel in that they for the first time demonstrate the ability of a brain region to compute subjective value across decision-making for self and other using a neural code that retains information across social referenced frames. Furthermore, our results extend the idea of a shared representation for value in the medial prefrontal cortex by indicating that this code in the dmPFC is generalizable across qualitatively different behavioral paradigms that divergently conceptualize subjective value.

## The similarity of subjective value representation across self and other

Our results are distinguished from many of those reported in previous studies by the level of similarity in subjective value coding across decision-making for self and other. There exists evidence for varying gradients in the medial prefrontal cortex, typically extending from ventral to more dorsal areas. Traditional thinking has implicated the vmPFC in self-referenced subjective valuation (*Kable and Glimcher, 2007*; *Levy et al., 2010*; *McClure et al., 2004*; *Behrens et al., 2008*; *Boorman et al., 2009*; *Knutson et al., 2005*), while the more dorsal areas of medial prefrontal cortex, typically referred to as the dmPFC – a functionally defined region that is overlapping with or proximal to anatomically defined areas such as the pre-SMA and dACC – are implicated more in social cognition (*Behrens et al., 2008*; *Frith and Frith, 2006*; *Hampton et al., 2008*). Corresponding to these findings, a recent approach using a prosocial learning framework has indicated a ventral to dorsal gradient in medial prefrontal cortex, with self value represented ventrally and other value represented dorsally (*Sul et al., 2015*). In our study, a cluster located in the anterior dmPFC was observed in a value-agnostic contrast of Other over Self trials in the intertemporal choice paradigm (*Figure 2*), and a cluster located more dorsally, anatomically proximal to the dACC and pre-SMA, was observed to more strongly signal other-referenced relative subjective value, although this effect was limited to the intertemporal choice paradigm (*Figure 3—figure supplement 1*).

Indeed, determining the functional specificity of the dorsal regions of medial prefrontal cortex in social cognition continues to be an active area of research. For example, a study using a reinforcement-learning task for self, other, or no one did not find evidence for any dorsal region of the medial prefrontal cortex in other-referenced choice, but instead found that the subgenual ACC corresponded to prediction errors specifically during learning for others and that the size of this effect was related to self-reported empathy scores (*Lockwood et al., 2016*). Additionally, the rostral ACC has been implicated in belief-based prediction errors during a competitive task (*Zhu et al., 2012*). Furthermore, the ACC gyrus subregion has been found across multiple studies to be vital in social cognition, signaling other-related information and motivation (*Apps et al., 2016*). In monkey studies utilizing single-unit recordings, the functionally defined dmPFC has been repeatedly implicated in social reward signaling (*Haroush and Williams, 2015*; *Noritake et al., 2018*). In our study, a region located within the dmPFC, particularly proximal to the dACC, signaled relative subjective value not only during Other trials, but also during Self trials, and it did so consistently across univariate and multivariate analyses as well as across tasks requiring different cognitive demands.

Notably, these results are in imperfect correspondence to an earlier paper which reported the vmPFC to be important in online subjective valuation and the dmPFC to be important in offline subjective valuation (*Nicolle et al., 2012*). While our paradigm is similar to the one used in this previous study, it is important to note that our univariate analyses were structured differently. When our analyses were structured as closely as possible to those detailed in the previous report (*Nicolle et al., 2012*), we were able to demonstrate a cluster in the posterior dmPFC that represented offline subjective value (*Figure 3—figure supplement 2*), roughly corresponding to previously reported findings. Additionally, while we were not able to definitively demonstrate encoding for offline relative subjective value in the dmPFC at the multivariate level, we did detect trends toward significant encoding of offline relative subjective value for both Self and Other trials (*Figure 4—figure supplement 1*; Self trials offline p=0.107, Other trials offline p=0.057, permutation test), lending support for conducting future research to more conclusively examine offline subjective value coding in the dmPFC at the multivariate level. Thus, while more research is, nevertheless, needed to fully consolidate earlier findings that indicate potential functional gradients in the medial prefrontal cortex (*Nicolle et al., 2012*; *Sul et al., 2015*), the current results reinforce the general notion that the dmPFC partakes in social cognition through distinct value-related computations.

## Activity patterns in the vmPFC and their role in signaling relative subjective value

Our results indicated that vmPFC activity corresponded more to relative subjective value than to the absolute value of the chosen option. This contradicts the notion detailed in some studies that vmPFC activity corresponds to the perceived value of the chosen option (*Wunderlich et al., 2010*; *Hare et al., 2009*). However, many other studies have indicated that activity in the medial prefrontal cortex corresponds more specifically to differences in subjective value between options

(*Nicolle et al., 2012*; *Boorman et al., 2009*; *FitzGerald et al., 2009*; *Lim et al., 2011*; *Grabenhorst and Rolls, 2009*; *Hare et al., 2011*; *Tremblay and Schultz, 1999*), consistent with our observed results.

Perhaps, more striking was the inability of classifiers trained using vmPFC activity to pass the decoding criterion of neural representation for relative subjective value when applying multivariate analysis techniques. Our results indicated that while the vmPFC signaled relative subjective value when voxels were treated as a homogenous group, the pattern of voxel activity within this region did not contain enough information to exceed an empirically derived null distribution (*Figure 4*). Indeed, multivariate approaches in conjunction with decision-making paradigms have not been used frequently. One previous study using MVPA in the frontal cortex to study the representation of value across reward categories found evidence for both category-dependent and category-independent value coding in varying subregions of the vmPFC (*McNamee et al., 2013*). More recently, representational similarity analysis (RSA), another multivariate analysis technique, was used to demonstrate that the vmPFC contained information regarding value univariately and information regarding category multivariately (*Zhang et al., 2017*), largely consistent with our results. Together, these findings indicate the importance of comparing univariate and multivariate activation patterns in decision-making paradigms to continue to explore the way by which the vmPFC and other frequently studied areas represent the neural correlates of value.

## The role of the dmPFC in computing relative subjective value

The role of the dmPFC and more specifically the anatomically proximal dACC subregion in value-based decision-making has been a point of contention (*Ebitz and Hayden, 2016*). Historically, the dmPFC and dACC have been thought to have a role not only in value comparison (*Kolling et al., 2016*; *Hare et al., 2011*; *Boorman et al., 2013*) but also in decision conflict (*Botvinick et al., 1999*). More recently, these opposing views for the role of the dmPFC and dACC have crystalized into two camps, one of which advocates for the importance of these regions in computing value across changing environments (*Kolling et al., 2016*; *Kolling et al., 2012*) while the other advocates for the importance of these regions in computing the expected value of control (*Shenhav et al., 2016*; *Shenhav et al., 2014*).

Our findings contribute to this argument by providing evidence for a specialized role of the dmPFC in common value representation across self and other. Specifically, our multivariate findings greatly add to our knowledge of dmPFC function. First, the dmPFC was the only of several medial prefrontal ROIs to decode relative subjective value, underscoring its importance in decision-making. Furthermore, our findings have essential implications for the common-currency hypothesis, which postulates that the brain continuously calculates value across varying categories and contexts to facilitate value comparison (*Levy and Glimcher, 2012*; *Bartra et al., 2013*). This hypothesis has mainly been attributed to vmPFC activity, as studies have shown overlapping value representation between categories in the vmPFC (*FitzGerald et al., 2009*; *Izuma et al., 2008*; *Chib et al., 2009*; *Levy and Glimcher, 2011*; *Sescousse et al., 2015*) as well as the ability of vmPFC activity patterns to generalize information regarding value across categories (*McNamee et al., 2013*; *Zhang et al., 2017*; *Gross et al., 2014*). Our vmPFC findings partially correspond to the previous literature, as our univariate analyses indicated an overlap not only between Self and Other trials, but also across tasks that involve different cognitive demands. However, our findings regarding the dmPFC extend the common-currency hypothesis to include other brain regions. Not only was there overlap in activity in the dmPFC related to relative subjective value across Self and Other trials and across behavioral paradigms, but our results also demonstrated that the neural code in the dmPFC that represents relative subjective value was able to retain information across both self and other, across univariate and multivariate levels of representation, and even across behavioral paradigms. These findings extend the reach of the common-currency hypothesis to include the dmPFC and its potential role in value comparison, using a code that computes subjective value information across agent and context differences.

Interestingly, the correlation that we observed with self-reported social attitudes indicated that this ability of the dmPFC to retain value information in its neural code across Self and Other trials may not be associated with prosocial attitudes, as one might expect. Instead, the opposite appears to be true, as increased agent cross-decoding accuracy was positively correlated with antisocial attitudes in our study (*Figure 6*). This indicates that a neural code in the dmPFC that is more separable

when making decisions for self versus other is associated with prosocial traits. While our sample size limited our ability to conclusively demonstrate this relationship, our findings raise the intriguing possibility that a neural code that makes individual-specific predications about self and other, perhaps by treating another individual as an independent agent, may underlie cognitive processes that result from an interactive process between prosocial preference and mentalizing about others.

That said, the tasks that we employed in our current study could not conclusively determine whether activity in the dmPFC signals subjective value or the expected value of control, nor were they designed to do so. When relative subjective value is low and dmPFC activity is heightened, this could have to do either with the intrinsic comparison of the value of the two options or with the amount of control necessary to make an accurate decision based on a given set of preferences. The fact that our dmPFC results could not be explained by response times suggests that our observed effects were not driven solely by trial difficulty. Yet, we do not believe that this finding is enough to invalidate the expected value of control hypothesis, especially when considering that the delay period of 5 s associated with each trial could decrease the extent of response time modulation by trial parameters. As such, the primary limitation of our study is that we are unable to conclusively differentiate between these opposing viewpoints.

## Conclusion

Both the vmPFC and dmPFC are components of broader networks that compute reward-related information across social and non-social contexts (*Fareri and Delgado, 2014*; *Smith et al., 2014*). Our study indicates that the code for relative subjective value in the dmPFC is robustly generalizable across various cognitive demands and social contexts in a way that is demonstrably related to social attitudes. This is in line with the common-currency hypothesis (*Levy and Glimcher, 2012*; *Bartra et al., 2013*) and other proposed theories suggesting how information may be retained across self and other (*Chang, 2013*). Our findings additionally argue for uniquely robust representation of relative subjective value in the dmPFC compared to other medial prefrontal regions. Together, these findings present the dmPFC as a nexus in computing variables relating to value-based decision-making, regardless of analytic technique used and across divergent tasks and social reference frames.

# Materials and methods

## Participants

Thirty participants provided written informed consent to take part in a study that took place over one to three sessions. The first session consisted only of a behavioral study using the intertemporal choice paradigm, while the second and third sessions consisted of an fMRI study using the intertemporal choice paradigm (*Figure 1a*) and risky choice paradigm (*Figure 1d*), respectively. Prior to the third session, participants completed a behavioral task online to gauge their self-referenced risk preferences. Nine of the 30 participants only attended the first session, 11 attended only the first and second sessions, and 10 attended all three sessions. Thirteen additional participants provided written informed consent and attended only the third session. All 30 initial participants were included in the analysis of preliminary intertemporal choice behavioral data from the first session (19 female, mean age 28.8, s.d. = 7.8). A total of 20 participants were included in behavioral and neural analyses of intertemporal choice acquired during the first fMRI session (15 female, mean age 29.1, s. d. = 8.7). Additionally, a total of 21 participants were included in behavioral and neural analyses of risk acquired during the second fMRI session (13 female, mean age 31.3, s.d. = 7.7). We based our individual sample sizes for both the intertemporal and risky choice tasks on previously published fMRI studies that utilized univariate (*Apps and Ramnani, 2017*) or multivariate (*McNamee et al., 2013*) analytical methods to explore the representation of subjective value. For the intertemporal choice dataset, one participant was excluded from further analyses due to excessive movement. For the risky choice dataset, one participant was excluded from further analyses due to excessive movement, and one participant was excluded from further analyses for failing to meet accuracy criterion in the task for learning the preferences of the other individual. All participants were between 18 and 55 years of age. Sex differences were not examined in this study due to a lack of the statistical power necessary to do so. The study was approved by the Yale School of Medicine Human

Investigation Committee (HIC #0910005795). All participants gave their informed consent prior to the experimental session and were paid for their participation.

## Stimuli and participant payment

All experiments were programmed in Matlab 2014a (MathWorks) using Psychtoolbox (version 3.0.12). For preliminary behavioral sessions and learning sessions, the task was presented on a computer monitor in a private behavioral testing room. Participants indicated their choices through button presses to a keyboard. For the preliminary intertemporal choice task, participants indicated their choice by releasing or continuing to hold a key. For the learning sessions, participants indicated their choice by pressing either a left or right key. For fMRI sessions, the task was projected to a screen at the back of the scanner, with a mirror used to allow participants to see the screen. Participants indicated their choices through left or right button presses to an MRI-safe button box in their right hand. Randomly selected trials for the purpose of payment were determined by rolling dice at the end of each experimental session. Funds were transferred to all participants using a web application called Square Cash in order to allow participants to be paid at the actual delays chosen in the intertemporal choice paradigm.

## Social intertemporal choice preliminary behavioral session

A preliminary behavioral session preceded the scanning session of the intertemporal choice task to evaluate the participants' baseline intertemporal preferences. In each trial, participants were required to choose between a delayed amount option in comparison to a common baseline offer of $10 immediately. Delays ranged from 1 to 180 days, while delayed amount options involved various amounts greater than $10 up to a maximum of $40. Participants made choices both for themselves ('Self' trials) and for another participant in the study ('Other' trials). For this behavioral session, participants were given no additional information about this other individual. One trial at random was selected from each of the participant's own Self trials and another participant's Other trials to contribute to the participant's payment. Trial types with every possible combination of delay and amount for both Self and Other constituted 84 trials, and every trial was repeated four times for a total of 336 trials. Task timing included a 1 s start screen, a 2 s presentation of the delayed option, a 2 s delay period, a 2 s choice period, and a 1 s confirmed choice period, followed by a 1–3 s intertrial interval. Task completion lasted approximately 60 min. Participants were paid a baseline amount of $10, plus added funds depending on their choice during one randomly determined Self trial as well as another participant's choice during one randomly determined Other trial. Funds were transferred to participants at the actual delay chosen during the experiment.

## Social intertemporal choice fMRI session

*Learning task.* Just prior to scanning, participants completed a learning task that allowed them to learn the preferences of another participant in the study for which they would be making choices. For this session, participants were matched with another individual with differing preferences, such that high-discounters were paired with low-discounters, and vice versa, so that self and other were dissimilar with respect to discounting preferences. Trial types were generated from a set of simulations and were unique to each individual participant. These trials were selected such that on at least 50% of trials each participant would prefer a different option than the individual whose preferences they were learning, as determined by each participant's preferences calculated from their choices in the preliminary behavioral session. For this task, participants had to indicate which of two options they thought the other individual in the study would choose. They were then shown the choice that the other individual would make on that particular trial and thus had the opportunity to iteratively learn the other individual's preferences. Delays ranged from 1 up to 180 days, with monetary amounts ranging from $0.01 up to $40. Each trial involved either a smaller amount of money sooner or a larger amount of money later. The task consisted of 80 trials completed over two blocks of 40 trials each. Task timing included a 2 s presentation of the options, a 3 s period in which the options remained on the screen and an arrow indicated the choice of the participant, a 2 s period in which the participant was able to view the choice of the other individual, and a 1 s inter-trial interval. Task completion lasted approximately 12 min. Accuracy was evaluated following the task, with a criterion

of 80% accuracy to be included in the analyzed dataset. No trials in the learning task directly impacted the payout of either the participant or the other individual.

*fMRI task.* This task involved participants making choices both for themselves and for the other participant whose preferences they just determined in the learning task and took place in the fMRI scanner. Trial types were generated uniquely for each participant such that the absolute difference between the subjective value of the two presented options, used to quantify relative subjective value, was uncorrelated between each participant and the other individual. These calculations were completed based on each participant's behavior during the preliminary behavioral session. Similar to the learning task, each trial involved an option of a smaller amount of money sooner or a larger amount of money later. Delays again ranged from 1 up to 180 days, and monetary amounts ranged from $0.01 up to $40. Participants were asked to either make choices for themselves during Self trials or the other individual during Other trials in alternating blocks. Importantly, participants were not explicitly asked to follow the other individual's preferences on Other trials, but simply told to make choices for that person howeverthey wished. The task consisted of 8 blocks of 30 trials each, with 15 of these being Self trials and 15 of these being Other trials counterbalanced for order. Participants therefore completed 240 trials in total, consisting of 120 Self trials and 120 Other trials. Each block also started with a dummy trial that was discarded in future analyses. Each Self or Other block began with an indication of the block type for 3 s. Task timing included a 5 s presentation of the options, a 2 s period in which the options remained on the screen and the participant was free to make their choice, a 2 s period in which the participant was able to view their choice, and a 3–7 s inter-trial interval (*Figure 1a*). Task completion lasted approximately 60 min. Participants were paid a baseline amount of $30 plus added funds depending on their choice during one randomly determined Self trial as well as another participant's choice during one randomly determined Other trial, with the decisions of high-discounters matched to those of low-discounters, and vice versa. Funds were transferred to participants at the actual delay chosen during the experiment.

## Social risky choice fMRI session

*Preliminary online choice task.* Unlike in the intertemporal choice paradigm, participant preferences were initially evaluated via a behavioral task administered online instead of with an in-person testing session. This task was delivered via Qualtrics prior to the learning and fMRI risky choice tasks. It consisted of 30 trials in which participants were asked to choose between a baseline option of a 50% chance of earning $5 or an alternative option with probabilities ranging from 13% up to 38% and monetary options ranging from $5 up to $65. All questions only asked participants about what they would prefer for themselves. To incentivize participants to answer accurately based on their preferences, one trial was randomly chosen at the conclusion of the fMRI session to be played out and contributed to total earnings.

*Learning task.* Similar to the intertemporal choice paradigm, participants completed a learning task just prior to scanning that allowed them to learn the preferences of another participant in the study for which they would be making choices. For this session, participants were matched with a fictional individual that had the same risk preferences that they had indicated in the preliminary online task. Participants were told that this was another actual participant in the study and were debriefed on the nature of this deception at the end of the session. Trial types were generated from a set of simulations and were unique to each individual participant. These trials were selected such that on approximately 50% of trials the fictional other individual would choose the riskier option, as determined by each participant's preferences calculated from their choices in the preliminary online task. For the learning task, participants had to indicate which of two options they thought the other individual in the study would choose, again similar to the intertemporal choice task. They were then shown the choice that the other individual would make on that particular trial and thus had the opportunity to iteratively learn these preferences. Probabilities of earning the monetary reward ranged from 20% up to 60%, with monetary amounts ranging from $0.01 up to $40. Each trial involved either a smaller amount of money with a larger probability or a larger amount of money with a smaller probability. The task consisted of 80 trials completed over two blocks of 40 trials each. Task timing included a 2 s presentation of the options, a 3 s period in which the options remained on the screen and an arrow indicated the choice of the participant, a 2 s period in which the participant was able to view the choice of the other individual, and a 1 s inter-trial interval. Task completion lasted approximately 12 min. Accuracy was evaluated following the task, with a criterion

of 80% accuracy to be included in the analyzed dataset. No trials in the learning task directly impacted the payout of either the participant or any other individual in the study.

*fMRI task.* Similar to the intertemporal choice task, the risky choice task involved participants making choices both for themselves and for the other participant whose preferences they just learned in the learning task and took place in the fMRI scanner. However, as mentioned above, the other participant in this task was fictional and was in fact based on the participant's own preferences as reported in the preliminary online task. Trial types were generated uniquely for each participant such that each participant would choose the riskier option on approximately 50% of trials. Again, these calculations were completed based on each participant's answers to the preliminary questionnaire. Similar to the learning task, each trial involved an option of a smaller amount of money with a larger probability or a larger amount of money with a smaller probability. Probabilities of earning the monetary reward again ranged from 20% up to 60%, and monetary amounts ranged from $0.01 up to $40. Participants were asked to either make choices for themselves during Self trials or the fictional other individual during Other trials in alternating blocks. Importantly, participants were not explicitly asked to follow the other individual's preferences on Other trials, but simply told to make choices for that person however they wished. The task consisted of 8 blocks of 30 trials each, with 15 of these being Self trials and 15 of these being Other trials counterbalanced for order. Participants therefore completed 240 trials in total, consisting of 120 Self trials and 120 Other trials. Each block also started with a dummy trial that was discarded in future analyses. Each Self or Other block began with an indication of the block type for 3 s. Task timing included a 5 s presentation of the options, a 2 s period in which the options remained on the screen and the participant was free to make their choice, a 2 s period in which the participant was able to view their choice, and a 3–7 s inter-trial interval (*Figure 1d*). Task completion lasted approximately 60 min. Participants were paid a baseline amount of $40, plus added funds depending on their choice during one randomly determined Self trial as well as one randomly determined Other trial. As mentioned above, one randomly determined choice in the preliminary online task was also chosen to potentially contribute to earnings. All selected options were then played out, such that participants had the stated chance at winning the amount of money indicated in the chosen option. Funds were transferred to participants immediately following the conclusion of the fMRI session, and participants were debriefed concerning any deception that occurred during the study.

## Acquisition of fMRI data

fMRI images were acquired at the Magnetic Resonance Research Center at Yale University on a 3.0 T Siemens Prisma MRI scanner using a 64-channel head coil. A high-resolution structural image was collected using a 3D MP-RAGE sequence (TR = 1900 ms, TE = 2.52 ms, flip angle = 9°, FOV = 350×263 mm, matrix = 256×96, slice thickness = 1 mm, 176 slices). Functional images were collected using a multiband sequence (TR = 2000 ms, TE = 33 ms, flip angle = 55°, FOV = 192×192 mm, matrix = 128×128, slice thickness = 1.5 mm, 81 slices, multiband acceleration factor = 3) with isotropic 1.5 $mm^3$ voxels.

## Preprocessing of fMRI data

Preprocessing was performed using the FMRIB Software Library (FSL, http://www.fmrib.ox.ac.uk/fsl). Images were skull-stripped using FSL's brain extraction tool (BET) with bias field and neck cleanup. The first five volumes (10 s) of each functional run were discarded to allow the MR signal to stabilize. Motion correction was performed using MCFLIRT linear realignment and a high pass filter of 0.01 Hz was applied. Data for whole-brain univariate general linear model (GLM) analysis were spatially smoothed at an FWHM of 5 mm. Data for multivariate multi-voxel pattern analysis (MVPA) were spatially smoothed at an FWHM of 2 mm. No slice timing correction was applied. Functional data were registered to high-resolution structural images using a 6 DOF transformation. For group level GLM analyses, images were normalized to the Montreal Neurological Institute template (2 mm MNI152).

## Self-reported measures of social attitudes

Social attitudes were assessed using Qualtrics for all participants who completed at least one fMRI session. Questionnaire links were sent after the final fMRI session, and participants were paid an additional $10 when all questionnaires were completed. Questionnaires included the Self-Report

Altruism Scale (*Rushton et al., 1981*), the Empathy Quotient (*Baron-Cohen and Wheelwright, 2004*), the Autism Quotient (*Baron-Cohen et al., 2001*), the Levenson Self-Report Psychopathy Scale (*Levenson et al., 1995*), and the Severity Measure for Social Anxiety Disorder (Social Phobia) – Adult (*APA, 2013*).

## Analysis of behavioral data

Behavioral data were analyzed using Matlab 2014a (MathWorks). All behavioral comparison tests consisted of two-tailed, nonparametric Wilcoxon signed rank or rank sum tests to a threshold value of p<0.05 with *N* defined as the number of participants, unless otherwise indicated. The *fminsearch* function in Matlab, which in turn uses the Nelder-Mead simplex algorithm, was used to estimate all model parameters.

### Intertemporal choice behavioral modeling

Decision-making behavior for self and other during the intertemporal choice task was modeled using a hyperbolic (*Equation 1*) decay function:

$$SVo = Ro * \frac{1}{1 + k * to}$$ (1)

This model assumes that the subjective value (*SV*) of each option (*o*) is determined by the monetary reward level (*R*) and the time to receive reward (*t*), depending on a participant-specific discounting parameter (*k*) that describes the steepness of each individual's devaluation of reward by time to reward. Previous studies have indicated that intertemporal choice is governed by a hyperbolic discounting model (*Kable and Glimcher, 2007*). We added variations of this model to our study, such that one model assumed a consistent value for *k* across self and other ($k_b$), while another model assumed distinct values for *k* for self ($k_s$) versus other ($k_o$). This approach was also taken in a recent study regarding the effects of social effort on subjective value (*Lockwood et al., 2017*). Finally, the softmax function (*Equation 2*) was used to calculate choice probability:

$$P(r) = \frac{e^{\beta*SVr}}{e^{\beta*SVr} + e^{\beta*SVl}}$$ (2)

In this equation, *P(r)* represents the probability of choosing the right option (*r*) with a subjective value of *SVr* as opposed to the left option (*l*) with a subjective value of *SVl*. A noise parameter (*β*) defines the stochasticity of each participant's choices. Models with either one *β* parameter across self and other or one *β* each for self and other were also examined. Thus, a total of four models were tested, including hyperbolic decay models with either one or two values for *k* and *β*. We used Bayesian information criterion (BIC) to compare model performance (*Schwarz, 1978*) in addition to using these BIC values to approximate a Bayes factor individually for each participant (*Wagenmakers, 2007*).

### Risky choice behavioral modeling

Decision-making behavior for self and other during the risky choice task was modeled using prospect theory (*Kahneman and Tversky, 1979*) (*Equation 3*):

$$SVo = (Po) * Ro^\alpha$$ (3)

This model assumes that the subjective value (*SV*) of a given option (*o*) is determined by the monetary reward level (*R*) and the probability of receiving reward (*P*), depending on a participant-specific risk preference factor (*α*) that describes the level of risk aversion of each individual. This approach has been used successfully to model risk behaviors in earlier studies (*Levy et al., 2010*). Similar to our modeling of the intertemporal choice task, we added a variation of this model to our study, such that one model assumed a consistent value for *α* across self and other ($\alpha_b$), while another model assumed distinct values for *α* for self ($\alpha_s$) versus other ($\alpha_o$). Again, models with either one *β* parameter across self and other or one *β* each for self and other were also examined. This resulted in a total of four models being tested. The softmax function (*Equation 2*) was used to calculate choice probability, and BIC was used to compare model performance (*Schwarz, 1978*) in addition to using these BIC values to approximate a Bayes factor individually for each participant (*Wagenmakers, 2007*).

## Quantification of subjective value

We conceptualized subjective value using two different approaches for neural analyses. First, we calculated the subjective value of the chosen option minus the subjective value of the unchosen option and noted this measure as relative subjective value. Next, we determined the subjective value of the chosen option alone. Thus, for further analyses, we conceptualized subjective value as either the relative subjective value – the signed difference between the chosen and unchosen options – or the absolute subjective value of the chosen option. For certain analyses, these values were used to separate trial types into high and low-value trials. This was accomplished by performing a median split, such that the half of trials with a low subjective value measure were considered 'low-value' trials, whereas the half of trials with a high subjective value measure were considered 'high-value' trials. To facilitate this categorization from continuous subjective value variables, trials were individually planned by design for each participant to maximize the intended separation of relative subjective value between high and low-value trials. While the actual distributions were not strictly bimodal as was initially intended due to changes in participant preferences between the preliminary and fMRI sessions, the mean and median of high-value versus low-value trials were still clearly separated by a minimum difference of 400%. The median split was performed for multivariate neural analyses to maximize the power of the utilized binary classification algorithms.

## Response time analysis

Response times for fMRI sessions were calculated as the difference in time between the onset of the choice period and the pressing of a button to indicate choice (*Figure 1a,d*). Response times were then compared between high and low-value trials as well as between Self and Other trials.

## Online versus offline analyses

Previous literature has indicated distinct neural correlates for the subjective value of the agent for whom a participant is making decisions (the online agent) versus the subjective value of the agent for whom a participant is not making decisions (the offline agent) (*Nicolle et al., 2012*). Thus, for models that involve distinct free parameters for self versus other-referenced decision-making, offline values can be calculated by switching the parameters used to calculate online values. Using the intertemporal choice paradigm as an example, using $k_s$ to calculate value during self-referenced decision-making constitutes online analyses, while using $k_o$ to calculate value during self-referenced decision-making constitutes offline analyses. For offline relative subjective value analyses, the chosen option was assumed to be the option with the higher subjective value. Online versus offline subjective value measures were uncoupled in the intertemporal choice paradigm by ensuring that the simulated trial set for each participant lead to uncorrelated subjective value measures across self and other.

## Definition of regions of interest (ROIs)

### Spherical ROIs

Spherical ROIs were produced for the ventromedial prefrontal cortex (vmPFC) (*Clithero and Rangel, 2014*) and dorsomedial prefrontal cortex (dmPFC) (*Venkatraman et al., 2009a*; *Venkatraman et al., 2009b*) using peak coordinates from previous studies and placed along the midline for MVPA analyses. One intermediate ROI (imPFC) was also placed along the medial prefrontal cortex between the vmPFC and dmPFC. A control ROI was placed in right somatosensory cortex (S1). Spherical ROIs were grown from a central coordinate to either 5 mm in diameter to extract effect sizes from univariate analyses or 8 mm in diameter for use in multivariate analyses (*Figure 4—source data 1*).

### Anatomical ROI

A bilateral anatomical ROI for the dmPFC was extracted from studies of macaque and human anatomical connectivity (*Neubert et al., 2015*; *Mars et al., 2016*). This dmPFC ROI corresponded to bilateral area 9 m and the posterior rostral cingulate zone (*Figure 4—source data 1*).

## Univariate analysis of fMRI data

Whole brain voxelwise regression analyses were conducted using the fMRI expert analysis tool (FEAT) in FSL. Regressors of interest were convolved with a gamma function. In all GLM analyses, the 5 s delay period before participants were instructed to indicate their decision was modeled. Six motion regressors were included in every model. Nuisance regressors were included for error trials and task block structure. All regressor values were mean-corrected and standard deviation-normalized prior to GLM analyses. Results for each participant were combined across eight runs using a fixed effects model. FSL's mixed effects Flame 1 and 2 model was used to create group averages, which were cluster corrected at a cluster-defining threshold of $p < 0.001$ uncorrected to satisfy $p < 0.05$ family-wise error (FWE). For ROI analyses, effect sizes were extracted and converted to percent signal change using FEATquery in FSL and compared using two-tailed, nonparametric Wilcoxon signed rank and Friedman tests to a threshold value of $p < 0.05$ with $N$ defined as the number of participants.

### GLM 1

Regressors of interest included Self trials and Other trials. Main effects as well as two contrasts, Self over Other and Other over Self, were modeled.

### GLM 2

Regressors of interest included online relative subjective value for self and online relative subjective value for other. Log of response time was included as a nuisance regressor. Main effects as well as two contrasts, online self relative subjective value over online other relative subjective value and online other relative subjective value over online self relative subjective value, were modeled.

### GLM 3

Regressors of interest included offline relative subjective value for self and offline relative subjective value for other. Log of response time was included as a nuisance regressor. Main effects were modeled.

### GLM 4

Regressors of interest included Self trial online subjective value of the chosen option, Self trial online subjective value of the unchosen option, Other trial online subjective value of the chosen option, Other trial online subjective value of the unchosen option, Self trial offline subjective value of the chosen option, Self trial offline subjective value of the unchosen option, Other trial offline subjective value of the chosen option, and Other trial offline subjective value of the unchosen option. Log of response time was included as a nuisance regressor. Contrasts of the offline subjective value of the chosen option over the offline subjective value of the unchosen option were modeled for Self and Other trials separately as well as Self and Other trials together. These methods were designed to recapitulate previous work (*Nicolle et al., 2012*) and were thus only applied to the intertemporal choice paradigm.

### GLM 5

Regressors of interest included subjective value of the chosen option for self and subjective value of the chosen option for other. Main effects were modeled.

### Neurosynth

A Neurosyth (*Yarkoni et al., 2011*) term search for 'social' was performed using forward-inference, with a threshold at $p < 0.01$ false discovery rate (FDR) corrected.

## Multivariate analysis of fMRI data

Spherical ROIs used to classify trial types included the vmPFC, dmPFC, S1 as a control region, as well as an intermediate ROI placed between the vmPFC and dmPFC in the medial prefrontal cortex. Analyses were also replicated using an anatomically defined ROI for the dmPFC. Individual beta weights for each 5 s decision-making period were obtained using AFNI's *3dDeconvolve* and

*3dREMLfit.* Patterns were mean-centered prior to performing pattern classification. Classification was performed using the *glmnet* model in the *caret* package in R (**Kuhn, 2016**) for each of the ROIs within each participant. This algorithm fits a generalized linear model under penalized maximum likelihood and contains two tuning parameters, alpha and lambda. The parameter alpha can be set between 0 ('ridge'), in which the elastic-net penalty shrinks the coefficients of correlated predictors toward each other, and 1 ('lasso'), in which the lasso tends to pick one of them and discard the others. The parameter lambda then controls the strength of the penalty. In our study, the parameter alpha was set outright to 0, and the model was optimized within each participant by allowing lambda to vary over a grid of 100. Nevertheless, we repeated all analyses with the parameter alpha set to 1, and adjustment of this parameter did not affect the qualitative pattern of results. To ensure that classifiers were never trained and tested on data from the same scanning run (**Mumford et al., 2014**), trials were split into four blocks. Classifiers were trained on data from three of these blocks and tested on the remaining block. All possible combinations of blocks were evaluated and averaged. The *downsampling* function in *caret* was used to correct for any situations in which labels were unbalanced. This procedure was used for all analyses, even those that involved training and testing classifiers on separate trials or datasets. Classification accuracy for the optimal model was obtained for each participant and then all participants were averaged to get the mean group accuracy for each ROI.

Significance was determined using an empirical null distribution for each ROI created by combining permutation and bootstrapping (**Lee and McCarthy, 2016**; **Stelzer et al., 2013**). For each ROI for each participant, training labels were permuted 100 times to create 100 datasets in which the label and the data were dissociated. Classification was then run as described above for each of these 100 permuted data sets to create 100 null accuracies for each participant. A group level null distribution for each ROI was then created by randomly sampling a null accuracy for each participant and averaging those accuracies across all participants 10,000 times. The true average group accuracy was then compared to the empirical null distribution for that ROI to determine the p-value for that accuracy. This procedure is critical for accurate determination of significance, as assumptions of a chance level of 50% are not valid in small sample sizes (**Combrisson and Jerbi, 2015**; **Golland and Fischl, 2003**). Direct comparisons between performance accuracy of classifiers trained to categorize different behavioral variables were performed using two-tailed, nonparametric Wilcoxon signed rank and Friedman tests to a threshold value of $p < 0.05$ with *N* defined as the number of participants.

## Classification of trial variables separately for self and other

Online relative subjective value, offline relative subjective value, and subjective value of the chosen option were placed into either 'high' or 'low' groups based on median split. Within Self trials and Other trials, pattern classifiers were trained to decode high versus low trials. Cross-validation ensured that classifiers were never trained and tested on data from the same run. Significance was determined via permutation test.

## Classification of online relative subjective value across self and other

Pattern classifiers trained on online relative subjective value from Self trials using the methods outlined above were tested on Other trials, and vice versa. This was done to determine the ability of the neural code to retain information for relative subjective value across self and other. Significance was determined via permutation test.

## Classification of online relative subjective value across tasks

Pattern classifiers trained on online relative subjective value from Self and Other trials during the intertemporal choice task were tested on Self and Other trials during the risky choice task, and vice versa. This was done to determine ability of the neural code to retain information for relative subjective value across behavioral paradigms. Significance was determined via permutation test.

## Correlations of multivariate analyses with self-reported data

We determined the correlation between the agent cross-decoding accuracy of classifiers to identify high versus low-value trials and self-reported social attitudes. Classifier cross-decoding accuracy was evaluated by averaging classifier accuracy when training on Self trials and testing on Other trials,

and vice versa. For participants who completed both the intertemporal and risky choice tasks, agent cross-decoding accuracy was averaged across both sessions. For self-reported data, a principal component analysis was performed on scores from all five questionnaires administered. The first principal component score for each participant was calculated and used as a measure of social attitudes. Correlation between relative subjective value agent cross-decoding accuracy and self-reported social attitudes was evaluated by Spearman's rank correlation to a threshold of p<0.05. Significance was confirmed via a permutation test in which corresponding axis labels were shuffled 10,000 times to create a null distribution. This analysis was then completed after decomposing agent cross-decoding into training on Self trials and testing on Other trials, and vice versa. Finally, this analysis was completed separately for the intertemporal and risky choice paradigms.

## Data visualization and determination of cluster locations

Statistical maps were visualized using FSL's FSLview. Anatomical locations of peak voxels for all clusters listed in source data files were determined according to the Harvard-Oxford cortical and subcortical atlases. Violin plots were generated in Matlab using the function *ksdensity* to perform kernel density estimations with a bandwidth of 0.025 commonly applied to all distributions.

## Data and code availability

Full neural datasets are available and downloadable from OpenNeuro. The Social Decision-Making Intertemporal Choice Task Dataset is here https://doi.org/10.18112/openneuro.ds001882.v1.0.5 and the Social Decision-Making Risky Choice Task Dataset is here https://doi.org/10.18112/openneuro.ds001883.v1.0.1. Matlab code for fitting models to behavioral data and R code for MVPA analyses are available and downloadable from GitHub (*Piva, 2019*; copy archived at https://github.com/elifesciences-publications/social-decision-making-fmri).

# Acknowledgements

We thank Patricia Lockwood and Matthew Apps for sharing behavioral analysis code and for useful comments on the project. We thank Spencer Birney for his help with data collection. Funded by a Natural Sciences and Engineering Research Council of Canada Fellowship (PGSD3-471313-2015) awarded to M.P.

# Additional information

## Funding

| Funder | Grant reference number | Author |
| --- | --- | --- |
| Natural Sciences and Engineering Research Council of Canada | PGSD3-471313-2015 | Matthew Piva |

The funders had no role in study design, data collection and interpretation, or the decision to submit the work for publication.

## Author contributions

Matthew Piva, Conceptualization, Resources, Data curation, Software, Formal analysis, Funding acquisition, Validation, Investigation, Visualization, Methodology, Writing—original draft, Project administration, Writing—review and editing; Kayla Velnoskey, Resources, Data curation, Software, Formal analysis, Methodology, Project administration; Ruonan Jia, Conceptualization, Resources, Methodology, Project administration; Amrita Nair, Resources, Investigation, Methodology, Project administration; Ifat Levy, Conceptualization, Resources, Formal analysis, Supervision, Investigation, Methodology, Writing—original draft, Project administration, Writing—review and editing; Steve WC Chang, Conceptualization, Resources, Data curation, Formal analysis, Supervision, Funding acquisition, Validation, Investigation, Methodology, Writing—original draft, Project administration, Writing—review and editing

### Author ORCIDs
Matthew Piva (iD) https://orcid.org/0000-0003-0906-9840
Steve WC Chang (iD) https://orcid.org/0000-0003-4160-7549

### Ethics

Human subjects: This study was approved by the Yale School of Medicine Human Investigation Committee (HIC #0910005795). All participants gave their informed consent prior to all experimental sessions and were paid for their participation.

### Decision letter and Author response

Decision letter https://doi.org/10.7554/eLife.44939.032
Author response https://doi.org/10.7554/eLife.44939.033

## Additional files

### Supplementary files

• Transparent reporting form
DOI: https://doi.org/10.7554/eLife.44939.026

### Data availability

Full neural datasets are available and downloadable from OpenNeuro. The Social Decision-Making Intertemporal Choice Task Dataset is here https://doi.org/10.18112/openneuro.ds001882.v1.0.5 and the Social Decision-Making Risky Choice Task Dataset is here https://doi.org/10.18112/openneuro.ds001883.v1.0.1. Matlab code for fitting models to behavioral data and R code for MVPA analyses are available and downloadable from GitHub (available at https://github.com/changlabneuro/social-decision-making-fmri; copy archived at https://github.com/elifesciences-publications/social-decision-making-fmri).

The following datasets were generated:

| Author(s) | Year | Dataset title | Dataset URL | Database and Identifier |
|---|---|---|---|---|
| Matthew Piva, Kayla Velnoskey, Ruonan Jia, Amrita Nair, Ifat Levy, Steve WC Chang | 2019 | Social Decision-Making Intertemporal Choice Task Dataset | https://dx.doi.org/10.18112/openneuro.ds001882.v1.0.5 | OpenNeuro, 10.18112/openneuro.ds001882.v1.0.5 |
| Matthew Piva, Kayla Velnoskey, Ruonan Jia, Amrita Nair, Ifat Levy, Steve WC Chang | 2019 | Social Decision-Making Risky Choice Task Dataset | https://dx.doi.org/10.18112/openneuro.ds001883.v1.0.1 | OpenNeuro, 10.18112/openneuro.ds001883.v1.0.1 |

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
