## [Decision Letter]

Thank you for submitting your article "The dorsomedial prefrontal cortex computes task-invariant relative subjective value for self and other" for consideration by *eLife*. Your article has been reviewed by three peer reviewers, one of whom is a member of our Board of Reviewing Editors, and the evaluation has been overseen by Michael Frank as the Senior Editor. The reviewers have opted to remain anonymous.

The reviewers have discussed the reviews with one another and the Reviewing Editor has drafted this decision to help you prepare a revised submission.

Summary:

This study examines common representations of subjective value in vmPFC and dmPFC, for two types of decision-making tasks (intertemporal and risky choices), and for decisions made with either the self or another individual. The emphasis is on regions such as the dmPFC that share a representation of relative subjective value across all combinations of conditions. This is an important goal, and the results support the notion that dmPFC activity encodes relative subjective value, regardless of whether the decision is made for oneself or another, and across decision task types.

Overall, there was a lot of enthusiasm for this paper. However, the reviewers also raised a number of major and minor concerns which need to be addressed. These condensed points are listed below.

Essential revisions:

1) All analyses are based on "relative subjective value" (RSV, |value left option – value right option|), defined as the absolute difference between the two choice options, rather than "relative chosen value", defined as the signed difference between the value of the chosen minus unchosen option (chosen value – unchosen value). Because the RSV signal is closely related to conflict or choice difficulty, there is a concern is that this signal might not identify brain areas involved in valuation per se, but more likely to identify areas involved in conflict or response selection, which are more likely to be shared across choice types even if the underlying valuation mechanism is different. Whereas chosen value – unchosen value is to some degree also correlated with choice difficulty, it is currently a reasonably well-established analysis, and it is the analysis of choice in most of the papers cited by the authors. The reviewers agreed that it would be important to reanalyze the data focusing on chosen value – unchosen value instead of RSV. In addition, the authors could focus on value sum (chosen + unchosen value), which is less related to conflict and difficulty. If those analyses failed to replicate previous studies, this could be still really interesting, and the authors may want to consider carefully why they do or do not replicate other previously reported effects.

2) Empirically determined chance null performance for all decoding analyses in which the classifier is trained and tested on the same trial types is systematically above 50%, indicating that the classifier is biased. This may be related to the fact that cross-validation was performed agnostic to run structure, or because of a potential non-independence in optimizing the lambda parameter. It would be important to reanalyze the data using a leave one run out cross-validation approach. If the median split is applied within each run, potential differences in mean value across runs should not affect classifier performance.

3) Please explore and report the behavioral data more thoroughly. For instance, it would be important to include plots depicting the relationship between value and choices for both self and others, indicating that value is indeed driving choices. Second, response times could be analyzed more comprehensively and displayed in appropriate graphs. For instance, were there differences in overall RT when deciding for self and others?

4) There are a number of issues with the correlation analysis of decoding accuracy and social attitudes that should be addressed. First, please compute these correlations without duplicating data points from the 10 subjects who participated in both sessions (for instance by averaging decoding accuracy in these subjects across both tasks). Second, please report how antisocial attitudes are related to behavioral performance, as one might expect that participants with more antisocial attitudes also show reduced correspondence between their decisions for the other person and the other person's preferences, or are less accurate when learning the other's preferences. Third, the authors use the "normalized" decoding accuracy, subtracting accuracy for self from accuracy for other. It might be interesting to know what is driving the effect, if they did not subtract one from the other.

5) It is possible that choice stochasticity differs between self and other trials. It would therefore be important to also test models that include separate beta parameters for self and other trials.

---

## [Author Response]

Essential revisions:1) All analyses are based on "relative subjective value" (RSV, |value left option – value right option|), defined as the absolute difference between the two choice options, rather than "relative chosen value", defined as the signed difference between the value of the chosen minus unchosen option (chosen value – unchosen value). Because the RSV signal is closely related to conflict or choice difficulty, there is a concern is that this signal might not identify brain areas involved in valuation per se, but more likely to identify areas involved in conflict or response selection, which are more likely to be shared across choice types even if the underlying valuation mechanism is different. Whereas chosen value – unchosen value is to some degree also correlated with choice difficulty, it is currently a reasonably well-established analysis, and it is the analysis of choice in most of the papers cited by the authors. The reviewers agreed that it would be important to reanalyze the data focusing on chosen value – unchosen value instead of RSV. In addition, the authors could focus on value sum (chosen + unchosen value), which is less related to conflict and difficulty. If those analyses failed to replicate previous studies, this could be still really interesting, and the authors may want to consider carefully why they do or do not replicate other previously reported effects.

We would like to thank the reviewers for bringing up this important issue. We agree that we conceptualized subjective value in a slightly different way than is common among previous papers in this field. To address this discrepancy, we have entirely replaced our original definition of relative subjective value to now mean “chosen value – unchosen value” throughout the whole paper, as requested. We have made this new definition clear in the Introduction, Results, and Materials and methods sections. We have applied the necessary changes to all relevant GLM and MVPA approaches. Importantly, all major findings hold following these changes. We believe that our paper now better corresponds to previous literature and examines a measure more commonly associated with subjective value, both of which represent major improvements to our manuscript.

We chose the “chosen value – unchosen value” option as opposed to the “chosen value + unchosen value” option provided by the reviewers for two main reasons. The first is that, to our knowledge, “chosen value – unchosen value” is a much more prevalent measure to use among other similar studies (for a comparable example, see Nicolle et al., 2012). Thus, our results are more comparable with the bulk of previous findings with this measure. Second, while “chosen value + unchosen value” is not directly related to the choices that participants make on any given trial, “chosen value – unchosen value” is intrinsically impacted by the choices that participants make. Therefore, if one wanted to examine measures most related to value-based choice behavior, “chosen value – unchosen value” might represent a better option. Furthermore, this logic also applies to our “value of the chosen option” analyses, which are also more intrinsically related to choice behavior than those associated with the “chosen value + unchosen value” measure.

Nicolle, A. *et al.* An agent independent axis for executed and modeled choice in medial prefrontal cortex. *Neuron*
**75**, 1114-1121, doi:10.1016/j.neuron.2012.07.023 (2012).

Please note however that we refer to the “chosen value – unchosen value” measure as “relative subjective value” in the updated manuscript. We believe that this is still a fair label for what we are reporting, as unlike the “chosen value + unchosen value” or “value of the chosen option” measures, the subjective value on each trial using this measure is subtractively normalized by whichever option the participant did not choose. Thus, although we have modified and clarified our definition of “relative subjective value” to be more robust and better correspond with previous literature, we refer to it using similar terminology as in the original manuscript (with clear indications that this is now defined as “chosen – unchosen value”).

Please see modified Figures 3, 4, 5, and 6; Figure 3—figure supplements 1-2, Figure 4—figure supplements 1-3, and Figure 6—figure supplements 1-2; Figure 3—source data 1, Figure 4—source data 1-2, and Figure 5—source data 1; and associated text for more detail on updated results using this new relative subjective value definition.

“A vital parameter in making decisions is the separation of the subjective value between the chosen and unchosen options, defined here as relative subjective value.”

“In order to determine the effect of subjective value on prefrontal activity, we calculated relative subjective value, defined as the subjective value of the chosen option minus the subjective value of the unchosen option.”

“We conceptualized subjective value using two different approaches for neural analyses. First, we calculated the subjective value of the chosen option minus the subjective value of the unchosen option and noted this measure as relative subjective value. Next, we determined the subjective value of the chosen option alone. Thus, for further analyses, we conceptualized subjective value as either the relative subjective value – the signed difference between the chosen and unchosen options – or the absolute subjective value of the chosen option.”

2) Empirically determined chance null performance for all decoding analyses in which the classifier is trained and tested on the same trial types is systematically above 50%, indicating that the classifier is biased. This may be related to the fact that cross-validation was performed agnostic to run structure, or because of a potential non-independence in optimizing the λ parameter. It would be important to reanalyze the data using a leave one run out cross-validation approach. If the median split is applied within each run, potential differences in mean value across runs should not affect classifier performance.

We appreciate the reviewers’ comments on this import methodological concern. We have now used the suggested leave one run out cross-validation approach in all relevant analyses, even those in which the null distribution means were already 50% in the original manuscript. We instantiated this by splitting the data up into four separate blocks. Classifiers were then trained on data from three of the four blocks and tested on data from the remaining block for all possible combinations. This was done both within Self and Other trials, across Self and Other trials, and even across the two distinct tasks. Thus, classifiers were never simultaneously trained and tested on trials from the same scanning run. We utilized tools in the *caret* package in R to deal with any imbalances in data sets. Specifically, we used the *downsampling* command to ensure that datasets were balanced. We thoroughly describe our new analytic approach in the Materials and methods section. Please note that we have also increased the size of our generated spherical ROIs from 5 to 8 mm. We found that these larger ROIs provided more stable decoding accuracies and are more in line with the size of ROIs typically used in MVPA approaches. Furthermore, we have simplified the intermediate ROIs placed between the vmPFC and dmPFC to one ROI that we now refer to as the intermediate medial prefrontal cortex (imPFC). We hope that this change streamlines our paper and provides fewer unnecessary comparisons that may confuse readers.

Following instantiation of our updated MVPA approach, the null distributions for all analyses are now centered closely on 50%, fully removing the mentioned bias. Please note that because we managed to fully remove the bias with this cross-validation approach, we continue to optimize the lambda parameter as in the original manuscript, as this is exceedingly common in instantiating the *glmnet* model that we used for these analyses.

Please see modified Figures 4, 5, and 6; Figure 4—figure supplements 1-3 and Figure 6—figure supplements 1-2; Figure 4—source data 1-2 and Figure 5—source data 1; and associated text for more detail on updated results:

“To ensure that classifiers were never trained and tested on data from the same scanning run^71^, trials were split into four blocks. Classifiers were trained on data from three of these blocks and tested on the remaining block. All possible combinations of blocks were evaluated and averaged. The *downsampling* function in *caret* was used to correct for any situations in which labels were unbalanced. This procedure was used for all analyses, even those that involved training and testing classifiers on separate trials or datasets.”

“Cross-validation ensured that classifiers were never trained and tested on data from the same run.”

3) Please explore and report the behavioral data more thoroughly. For instance, it would be important to include plots depicting the relationship between value and choices for both self and others, indicating that value is indeed driving choices. Second, response times could be analyzed more comprehensively and displayed in appropriate graphs. For instance, were there differences in overall RT when deciding for self and others?

We thank the reviewers for the opportunity to report the findings of our behavioral analyses more fully. First, we have now reported behavioral findings from the preliminary intertemporal choice task that we used to asses participants’ baseline preferences for self and other as well as the learning tasks for the intertemporal choice and risky choice tasks in which participants learned about the other person for whom they would be making choices (Figure 1—figure supplement 1). These findings include example discounting curves from the preliminary intertemporal choice data for two participants as well as group-averaged learning curves from the intertemporal and risky choice learning tasks.

Second, we have now demonstrated that subjective value explains participant decisions better than monetary value or delay alone in the intertemporal choice task for Self and Other trials (Figure 1—figure supplement 2A; versus Self monetary, *z* = 3.72, *P* < 0.001, vs Self delay, *z* = 3.92, *P* < 0.001, vs Other monetary, *z* = 3.85, *P* < 0.001, vs Other delay, *z* = 4.01, *P* < 0.001, Wilcoxon signed rank test). Furthermore, we have also now demonstrated that subjective value explains participant decisions better than monetary value or probability of receiving reward alone in the risky choice task for Self and Other trials (Figure 1—figure supplement 2B; vs Self monetary, *z* = 3.91, *P* < 0.001, vs Self probability, *z* = 4.01, *P* < 0.001, vs Other monetary, *z* = 4.01, *P* < 0.001, vs Other probability, *z* = 4.01, *P* < 0.001, Wilcoxon signed rank test). Together, these results indicate that participants’ choices were guided by subjective value and not necessarily by solely the monetary value or the delay to or probability of receiving a certain reward:

“We next aimed to determine whether the subjective value of each option on a given trial calculated using the best fitting models for each paradigm accounted for choice behavior. To demonstrate that participants were taking subjective value into account when making choices, we calculated the proportion of trials in which participants chose the option with the higher subjective value, monetary value, or either lower delay for the intertemporal choice task or higher probability for the risky choice task. As anticipated, participants made choices according to subjective value significantly more than either monetary value or delay alone for both Self and Other trials in the intertemporal choice task (Figure 1—figure supplement 2A; compared to Self monetary, *z* = 3.72, *P* < 0.001, Self delay, *z* = 3.92, *P* < 0.001, Other monetary, *z* = 3.85, *P* < 0.001, Other delay, *z* = 4.01, *P* < 0.001, Wilcoxon signed rank test). Similarly, participants made choices according to subjective value significantly more than either monetary value or probability of reward for both Self and Other trials in the risky choice task (Figure 1—figure supplement 2B; compared to Self monetary, *z* = 3.91, *P* < 0.001, Self probability, *z* = 4.01, *P* < 0.001, Other monetary, *z* = 4.01, *P* < 0.001, Other probability, *z* = 4.01, *P* < 0.001, Wilcoxon signed rank test).”

Finally, we examined response times in relation to two variables: low versus high-relative subjective value trials (using the new chosen – unchosen definition), as well as Self versus Other trials. For both the intertemporal and risky choice tasks, response time was higher for low-relative subjective value trials, indicating that the relative value of the two presented options did impact response time behavior (Figure 1—figure supplement 3A; intertemporal *z* = 3.10, *P* = 0.002, risk *z* = 2.38, *P* = 0.017, Wilcoxon signed rank test). However, no difference was observed between Self and Other trials (Figure 1—figure supplement 3B; intertemporal *z* = 1.01, *P* = 0.313, risk *z* = 1.20, *P* = 0.231, Wilcoxon signed rank test):

“Finally, we examined response times in relation to the subjective value associated with each trial as well as whether participants were making decisions for themselves or another individual. For the subjective value analysis, we first calculated the relative subjective value of each trial by subtracting the value of the unchosen option from the value of the chosen option. We then median split trials into high and low relative subjective value. Response times were slower for trials with lower relative subjective value than trials with higher relative subjective value for both the intertemporal and risky choice paradigms (Figure 1—figure supplement 3A; intertemporal *z* = 3.10, *P* = 0.002, risk *z* = 2.38, *P* = 0.017, Wilcoxon signed rank test). However, no differences in response times were observed between Self trials and Other trials in either paradigm (Figure 1—figure supplement 3B; intertemporal *z* = 1.01, *P* = 0.313, risk *z* = 1.20, *P* = 0.231, Wilcoxon signed rank test).”

We believe that these additional analyses represent a much more thorough examination of our behavioral dataset and greatly improve our manuscript compared to the initial version.

4) There are a number of issues with the correlation analysis of decoding accuracy and social attitudes that should be addressed. First, please compute these correlations without duplicating data points from the 10 subjects who participated in both sessions (for instance by averaging decoding accuracy in these subjects across both tasks). Second, please report how antisocial attitudes are related to behavioral performance, as one might expect that participants with more antisocial attitudes also show reduced correspondence between their decisions for the other person and the other person's preferences, or are less accurate when learning the other's preferences. Third, the authors use the "normalized" decoding accuracy, subtracting accuracy for self from accuracy for other. It might be interesting to know what is driving the effect, if they did not subtract one from the other.

We thank the reviewers for these comments and entirely agree with these concerns. We now calculate all correlations without duplicating data points from the 10 participants who completed both sessions by averaging decoding accuracy in these participants across both tasks, as suggested by reviewers. The one exception to this is for the behavioral accuracy analyses detailed below. We determined accuracy in these analyses by calculating the deviation of each participant’s best-fitting parameter from the actual parameter of the other participant. Because these parameters were of at least an order of magnitude different across tasks, we decided to evaluate correlations separately across tasks in this case.

Importantly, the results that we reported in our original manuscript did not remain significant after using our new definition of relative subjective value (chosen – unchosen value) along with our new cross-validation approach to MVPA as well as with the suggestion to average participants who attended both sessions. While the correlation was still in the negative direction, the effect was far below significance (Author response image 1; *r* = -0.112, *P* = 0.547, Spearman correlation).

**Author response image 1. respfig1:** Correlation between dmPFC accuracy in Other trials minus Self trials and self-reported social attitudes principal component score.

As the reviewers also suggested, we examined dmPFC decoding accuracy during Other trials without normalizing to Self trials. Again, no correlation was observed (Author response image 2; *r* = 0.095, *P* = 0.612, Spearman correlation).

**Author response image 2. respfig2:** Correlation between dmPFC accuracy in Other trials and self-reported social attitudes principal component score.

Next, we looked at the relationship between social attitudes and how accurate participants were at estimating the preferences of the other participant for whom they were making choices. As mentioned above, we quantified this as the deviation of each participant’s best-fitting parameter from the actual parameter of the other participant. Again, no correlation was observed for the intertemporal choice task (Author response image 3; *r* = -0.146, *P* = 0.538, Spearman correlation) or for the risky choice task (Author response image 4; *r* = -0.171, *P* = 0.456, Spearman correlation). Noise in both the behavioral choice data as well as in measures of social attitudes could contribute to our inability to determine a clear correlation between these variables. Furthermore, it need not be a distinct signature of prosocial attitudes to faithfully replicate another individual’s preferences. For example, in the intertemporal choice task, a very patient individual paired with a very impatient individual may act to make more patient choices for the other person as a way of shielding this other person from what the more patient individual perceives as a suboptimal decision.

**Author response image 3. respfig3:** Correlation between correspondence of choice behavior with partner preferences and self-reported social attitudes principal component score for the intertemporal choice task.

**Author response image 4. respfig4:** Correlation between correspondence of choice behavior with partner preferences and self-reported social attitudes principal component score for the risky choice task.

Finally, we performed an analysis that was suggested by reviewer #1 in their individual comments. This suggestion was to evaluate a potential correlation between social attitudes and the cross-decoding accuracy when training on Self trials and testing on Other trials. We found the prospect of this analysis intriguing but first performed it with one modification. Cross-decoding accuracy, rather than being a measure of accurate neural representation of value per se, is more a measure of how distinct two models are from one another. Two very similar coding schemes would likely lead to high cross-decoding accuracy, while two more dissimilar schemes would likely lead to lower cross-decoding accuracy. Therefore, the accuracy when training on Other trials and testing on Self trials could be just as important for this analysis.

Following reviewer #1’s valuable insight, we thus calculated the cross-decoding accuracy for all participants, averaging accuracy when training on Self trials and testing on Other trials, and vice versa. This analysis yielded a significant positive correlation for agent cross-decoding accuracy in the dmPFC and antisocial attitudes (Figure 6C; *r*= 0.40, *P* = 0.025, Spearman correlation; *P* < 0.05 confirmed via permutation test). Importantly, such a correlation was not observed in the vmPFC (Figure 6B; *r*= -0.08, *P* = 0.895, Spearman correlation). At first, these results seem counterintuitive. How could higher cross-decoding accuracy be associated with higher antisocial attitudes, as is suggested by the positive direction of the correlation? As we mentioned previously, it is important to consider that cross-decoding accuracy is not so much a measure of how well one’s dmPFC represents subjective value, but instead how similar the code for subjective value is across self and other agents in our study. Cross-decoding accuracy is therefore likely more of a measure of how generalizable classifiers are across agents. If a classifier is not able to generalize well across multiple datasets, these datasets probably utilize differentiated coding for subjective value. When these results are approached from this perspective, they start to become more interpretable and even intuitive. Participants who are more prosocial display dmPFC subjective value classifiers that can be more differentiated between self and other, demonstrating a more distinct code when making choices for other as opposed to self. Conversely, participants who are more antisocial display dmPFC subjective value classifiers that are less able to differentiate, indicating a more similar code for self and other value-based decision-making. It therefore seems that the significant cross-decoding that we observe in our study might not be a measure of prosocial empathy, as one might expect, but instead a potential measure of antisocial apathy in which participants make decisions for themselves and others using an indistinct neural code that does not distinguish other from self.

To ensure that this correlation was not simply observed by chance, we decided to also examine correlations between social attitudes and cross-decoding accuracy separately for training on Self trials and testing on Other trials, and vice versa. Results for the vmPFC remained negative (Figure 6—figure supplement 1A; train Self test Other *r* = 0.03, *P* = 0.880, train Other test Self *r* = -0.08, *P* = 0.653, Spearman correlation). However, the correlation between social attitudes and agent cross-decoding accuracy in the dmPFC remained remarkably intact in the same direction, with a significant relationship observed for training on Self and testing on Other (Figure 6—figure supplement 1B left; *r* = 0.453, *P* = 0.011) and a strong trend observed for training on Other and testing on Self (Figure 6—figure supplement 1B right; *r* = 0.341, *P* = 0.061). These results indicate that the correlation we now report is robust.

Finally, reviewer #1 also noted that they wanted to see the correlations individually for the intertemporal and risky choice tasks. When we performed this analysis separately for each task, we still observed no correlation in the vmPFC (Figure 6—figure supplement 2A; intertemporal *r* = -0.147, *P* = 0.537, risk *r* = -0.040, *P* = 0.862, Spearman correlation). However, even with diminished statistical power from reduced sample size, we still observed a trend toward a positive correlation in both tasks in the dmPFC (Figure 6—figure supplement 2B; intertemporal *r* = 0.415, *P* = 0.070, risk *r* = 0.423, *P* = 0.056, Spearman correlation). These results indicate that the observed relationship was not necessarily driven by the similarity or dissimilarity of the other participant or by specific task demands.

We thank the reviewers for the opportunity to revisit these analyses. We feel that these updated results are more nuanced and greatly add to our manuscript. These results are further discussed in the manuscript in the sections copied below:

“To determine the relationship between this measure and agent cross-decoding of relative subjective value across Self and Other trials, we averaged encoding of value when training on Self trials and testing on Other trials, and vice versa, in either the vmPFC or dmPFC. […] As mentioned above, since agent cross-decoding is primarily a measure of how similar value coding is for Self and Other trials in the dmPFC, these results indicate that participants who report more antisocial attitudes tend to have value coding that is less differentiated when making decisions for themselves as opposed to other individuals in our experimental paradigms.”

“Interestingly, the correlation that we observe with self-reported social attitudes indicates that this ability of the dmPFC to retain value information in its neural code across Self and Other trials may not be associated with prosocial attitudes, as one might expect. Instead, the opposite appears to be true, as increased agent cross-decoding accuracy is positively correlated with antisocial attitudes in our study (Figure 6). This indicates that a neural code in the dmPFC that is more separable when making decisions for self versus other is associated with prosocial traits. While our sample size limits our ability to conclusively demonstrate this relationship, our findings raise the intriguing possibility that a neural code that makes individual-specific predications about self and other, perhaps by treating another individual as an independent agent, may underlie cognitive processes that result from an interactive process between prosocial preference and mentalizing about others.”

5) It is possible that choice stochasticity differs between self and other trials. It would therefore be important to also test models that include separate beta parameters for self and other trials.

We thank the reviewers for the opportunity to explore our modeling more thoroughly. We have now included models that have two beta parameters, one for Self trials and one for Other trials, in our model testing. In both the intertemporal and risky choice tasks, models with one beta parameter for Self and Other trials fit best according to BIC (Figure 1B, E). We therefore continued to calculate the subjective value of each option using the same modeling that we used in the original manuscript.